SOFTWARE

# Centrifuger: lossless compression of microbial genomes for efficient and accurate metagenomic sequence classification

Li Song[1,2,3]* and Ben Langmead[4]

*Correspondence:
Li.Song@dartmouth.edu

[1] Department of Biomedical Data Science, Dartmouth College, Hanover, NH, USA
[2] Department of Computer Science, Dartmouth College, Hanover, NH, USA
[3] Department of Microbiology and Immunology, Dartmouth College, Hanover, NH, USA
[4] Department of Computer Science, Johns Hopkins University, Baltimore, MD, USA

## Abstract

Centrifuger is an efficient taxonomic classification method that compares sequencing reads against a microbial genome database. In Centrifuger, the Burrows-Wheeler transformed genome sequences are losslessly compressed using a novel scheme called run-block compression. Run-block compression achieves sublinear space complexity and is effective at compressing diverse microbial databases like RefSeq while supporting fast rank queries. Combining this compression method with other strategies for compacting the Ferragina-Manzini (FM) index, Centrifuger reduces the memory footprint by half compared to other FM-index-based approaches. Furthermore, the lossless compression and the unconstrained match length help Centrifuger achieve greater accuracy than competing methods at lower taxonomic levels.

**Keywords:** FM-index, r-index, Metagenomic, Compact data structure

## Background

Metagenomic sequencing enables comprehensive profiling of microbiomes in a sample and has been widely applied to study natural environments [1, 2], infectious diseases [3], allergies [4], and cancers [5]. Taxonomic classification labels each sequencing read with taxonomy IDs representing its most likely taxon of origin. This has become an important step in translating raw sequencing data into meaningful microbiome profiles [6]. Classification is usually conducted by comparing the read sequence to all the sequences in a database of microbial reference genomes, such as RefSeq [7], Gene-Bank [8], or GTDB [9]. The growth of available microbial reference genomes creates a strong need for memory-efficient structures. Many methods turn to lossy representations of the database. For example, Kraken2 [10] reduces the space by storing minimizers [11] instead of all the k-mers as in Kraken [12]. Other approaches, such as MetaPhlAn [13, 14] and CLARK [15], build the database out of only a selected subset of sequences, i.e., marker genes or discriminative k-mers. Ganon [16] and KMCP [17]

utilize probabilistic data structures that discard k-mer identity but support checking k-mer presence with false positive probability. While these strategies reduce the memory requirement, they lose valuable sequence information, which may lower accuracy when classifying read to lower taxonomic levels. We previously co-developed the taxonomic classification method Centrifuge [18] that used the memory-efficient Burrows-Wheeler transformed (BWT) sequence [19] and the Ferragina-Manzini (FM) index [20]. Centrifuge searches for semi-maximal matches with no length constraints, avoiding the decreased taxonomic specificity of k-mers when the genome database is large [21]. However, the FM-index grows linearly with the database size and the lossy compression strategy proposed in Centrifuge is not scalable, making Centrifuge less usable in the context of large and growing genome databases.

Related genomes share similar sequences, giving genome databases a degree of repetitiveness. Lempel–Ziv family indexes [22], context-free grammars [23], run-length compressed BWT indexes (RLBWT) [24], and the move structure [25] exploit this repetitiveness to reduce index size losslessly while supporting efficient search queries. For example, r-index [26] builds upon the RLBWT and fits in $O(r)$ words, where $r$ is the number of runs in the BWT sequence. The FM-index, by contrast, usually requires $O(n)$ words, where $n$ is the size of the database and is also the length of the BWT sequence.

While $O(r)$-space methods achieve good compression for highly repetitive sequences such as collections of human genomes, microbial genomes are more diverse. Applying these compact representations may take more space than the uncompressed wavelet tree [27]. Therefore, we designed two compact data structures, called run-block compressed BWT (RBBWT) and hybrid run-length compressed BWT, to effectively compress the BWT sequence for the intermediate level of repetitiveness characteristic of microbial genome databases. RBBWT achieved the best overall performance in both time and space efficiency when compared to other compression methods. Inspired by this observation, we developed the software tool Centrifuger (Centrifuge with RBBWT), which rapidly assigned the taxonomy IDs for a sequencing read while consuming half the memory of a conventional FM-index.

## Results

### Method overview

Centrifuger assigns a taxonomic ID to each input read or read pair by searching against a losslessly compressed FM-index built from a microbial genome database (Fig. 1, Methods). The FM-index contains two memory-consuming components, the data structure supporting rank queries over the BWT sequence, and the sampled suffix array. We propose a novel compact structure, the run-block compressed sequence, to reduce the size needed to store the BWT sequence. For the sampled suffix array, we save space by storing only the sequence ID for each sampled position on the BWT sequence, omitting information about the offset within the genome. The classification algorithm scans the read twice, once for the original sequence and once for the reverse-complement sequence. Each scan looks for semi-maximal matches, by repeatedly extending the match with the backward search until reaching a mismatch, then skipping the base immediately after the point where the backward search terminates. We call the match semi-maximal because only one end of the match cannot be extended further. For each

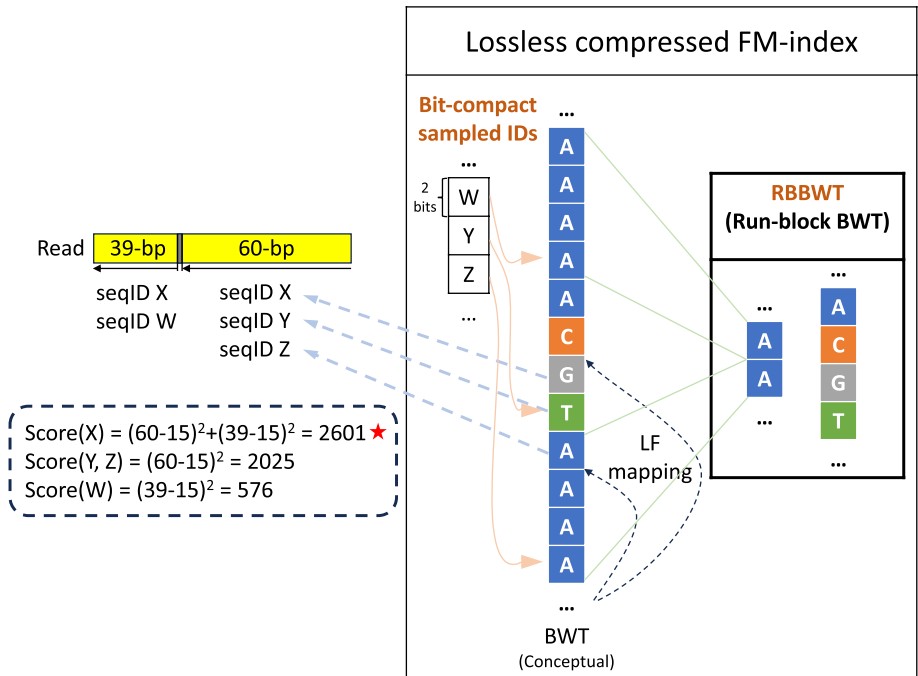

**Fig. 1** Overview of Centrifuger. Left: classification procedure on the forward read. Centrifuger searches from the end of the read and applies the backward search to extend the match until reaching a mismatch. This yields the first 60-bp exact match hitting three sequences {X, Y, Z} in the database. Centrifuger then skips the mismatch and restarts the search again, giving the second 39-bp match hitting two sequences {X, Y}. The same search procedure applies to the reverse complement of the read. Centrifuger then scores each matched sequence and classifies the read to the sequences with the highest scores, where the example read is classified to the sequence X with the score 2601. Right: the structure of Centrifuger's lossless compressed FM-index. Centrifuger utilizes the RBBWT representation for the BWT sequence. In the example of compressing the BWT sequence "AAAAACGTAAAA", RBBWT represents it as two sequences "AA" and "ACGT" when the block size is 4. For the sequence IDs that are sampled on the BWT sequence, Centrifuger will compact their bits representation. In this example, there are four sequences in the database (W, X, Y, Z), so 2 bits are sufficient to represent the sequence ID. Therefore, for the substring of the BWT sequence shown in the example, Centrifuger spends 6 bits to represent sequence IDs that are sampled every other four positions on the BWT sequence

match, Centrifuger retrieves the sequence IDs associated with entries in the matching BWT interval. Centrifuger adds a score, which is a quadratic function of the match length, for each retrieved sequence ID. After all the valid matches are processed, the highest-scoring taxonomy IDs translated from the sequence IDs are reported as the classification result. When the number of reported IDs exceeds the user-specified threshold (default report threshold 1), Centrifuger reduces the number to within the threshold by promoting some taxonomy IDs to their lowest common ancestors (LCAs) in the taxonomy tree. In other words, Centrifuger reports the LCA of the taxonomy IDs for a read by default, as in Kraken2.

### The computational efficiency of run-block compression

Run-block compressed sequence is a novel compact data structure that achieves sublinear space usage both in theory ($O\left(\frac{n}{\sqrt{l}}log\sigma\right)$ bits. $l$: average run length, i.e., $\frac{n}{r}$; $\sigma$: alphabet size) and practice. Centrifuger applies run-block compression to reduce the size of the BWT sequence, yielding the RBBWT: Run-Block compressed BWT. Rank

queries on the RBBWT, which form the basis for LF-mapping in the backward search, are also highly efficient, having a time complexity of $O(log\sigma)$ (more information in the "Methods" section).

We compared RBBWT with three other representations of BWT sequences: the standard wavelet tree, the RLBWT as implemented in the r-index package [28] using sdsl library [29], and the hybrid run-length compression (Methods). We measured the change in space usage when adding non-plasmid sequences from the species *Escherichia fergusonii* (taxonomy ID 564) to the structure. While the wavelet tree grew linearly as more genomes were added, RBBWT, RLBWT and its hybrid version grew more slowly (Additional File 1: Fig. S1A). When there was little repetitiveness in the genomes, RBBWT and hybrid run-length compression took almost the same amount of space as the wavelet tree. From another perspective, when the average run length of the BWT increased, the number of bits to represent a nucleotide in the wavelet tree remained constant (0.31 bits/bp in our implementation), and the other three compression methods needed fewer bits (Fig. 2A). RBBWT consumed the least or similar space compared with the run-length-based compression methods when $l$ was less than 10. When the BWT was constructed from all the genomes under taxonomy ID 564 with $l$ equaling 18.8, RBBWT was still small, consuming 57.8% less space than the wavelet tree and 29.8% more space than RLBWT. We also compared the space usage of the BWT representations by adding the genomes from the species *Chlamydia trachomatis* (taxonomy ID 810) whose strains had highly similar sequences [18]. Again, RBBWT was the most memory-efficient data structure when $l$ was less than or around 10 (Additional File 1: Fig. S2A). For this species, $l$ reached 56.0 after adding all the genomes, and RBBWT's space was about a quarter of the uncompressed wavelet tree's and twice as much as RLBWT's in this case.

We next compared the space usage when compressing the BWT sequence for genomes from the same genus. We examined the genus *Legionella* (taxonomy ID 445) containing 150 genomes. Since genomes from the same genus were more diverse than genomes from the same species, the final $l$ was 7.1 after adding all the non-plasmid sequences. RBBWT consumed the least memory among the benchmarked compression methods, using 46.9%, 24.8%, and 2.3% less space than wavelet tree, RLBWT, and hybrid run-length compressed BWT, respectively, in the end (Fig. 2B and Additional file 1: Fig. S1B).

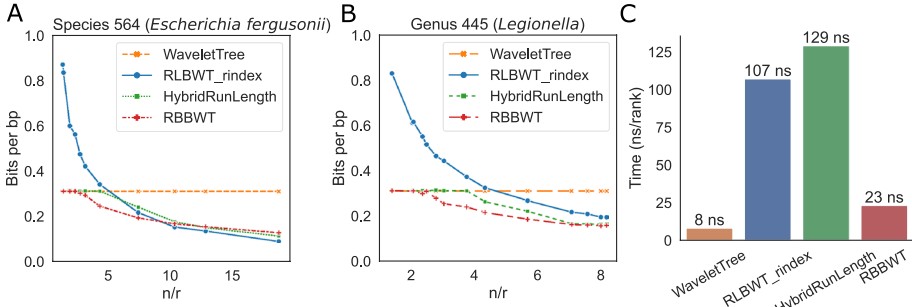

**Fig. 2** Computational efficiency of the wavelet tree, RLBWT, hybrid run-length compression, and RBBWT. **A** Bits used to represent one base pair (bp) as the average run length of the BWT sequence ($n/r$) increases when representing increasingly more genomes with species ID 564 (*Escherichia fergusonii*). **B** Bits used to represent one bp when representing genomes with genus ID 445 (*Legionella*). **C** Rank query time

A similar trend was observed when conducting the experiment on the genomes from the genus *Chlamydia* (taxonomy ID 810, Additional file 1: Fig. S2B). We further compared the speed of rank queries by averaging the time for finding the rank of 'A' for each of the first ten million positions in the BWT sequence of all the *Legionella*'s genomes, i.e., the average time of calling rank$_{A'}$(1, BWT) to rank$_{A'}$(10,000,000, BWT). Rank query in RBBWT was about five times faster than in RLBWT and only three times slower than using a wavelet tree (Fig. 2C). Hybrid run-length compression was the slowest method. In both the species ID 564 and the genus ID 445 analysis, the block size of RBBWT was automatically determined to be 8 (Methods) after adding all the genomes, supporting the mild repetitiveness in the microbial genome database. To summarize, RBBWT is more memory-efficient and supports faster rank queries compared to RLBWT when compressing microbial genomes.

## Performance on classifying simulated data

We compared Centrifuger, Centrifuge, Kraken2, Ganon, and KMCP's accuracy on one million 100-base-pair (bp) paired-end short reads simulated by Mason [30] from 34,190 prokaryotic complete genomes (RefSeq bacteria + archaea). We set the sequencing error rate in Mason to be 1%, a value that was higher than the Illumina sequencing platform, to model the microbial genome variations in real data. All five methods built the database indices on the same set of genomes. The average run length of the BWT was about 6.8, and the block size of RBBWT was automatically determined to be 8. We used TP (true positive) for the number of reads that are correctly classified at the specified taxonomy node or in its subtree, T (true) for the number of input reads, and P (positive) for the number of reads that were correctly classified at the specified taxonomy level or below. Therefore, we define the sensitivity as $\frac{TP}{T}$, and precision as $\frac{TP}{P}$. The strain-level classification evaluation in each method was for the reads classified to leaf nodes in the taxonomy tree. On this simulated data set, Centrifuger achieved the best accuracy at all taxonomy levels and was significantly better than other methods at species and genus levels (Fig. 3A, Additional file 1: Table S1). For example, Centrifuger's sensitivity was 34.5% higher than both Centrifuge and Kraken2, 20.0% higher than Ganon, and 116.0% higher than KMCP at the species level. All five classifiers had comparable precision except at the strain level. Centrifuge's low sensitivity could be due to its policy of not resolving taxonomy IDs for matches hitting too many places in the database, where the threshold for the number of hits of a match was 40·report_threshold (default value of Centrifuge's report threshold for the taxonomy IDs is 5). The strategy of handling matches on repetitive regions was one of the main differences between Centrifuger and Centrifuge during the classification stage (Methods). In addition to Mason, we compared the five classifiers on another set of one million 100-bp paired-end short reads simulated by ART [31], where the error rate was set to ART's default value (about 0.15%). All methods produced similar accuracy as the Mason-generated data. Centrifuger still achieved significantly higher sensitivity at the species and genus level while having comparable precision as the other methods (Additional file 1: Fig. S3). In both simulated data sets, the sensitivity at the strain level was very low (< 25%) for all five methods, suggesting that most reads cannot be uniquely assigned to a strain in the RefSeq prokaryotic genome database.

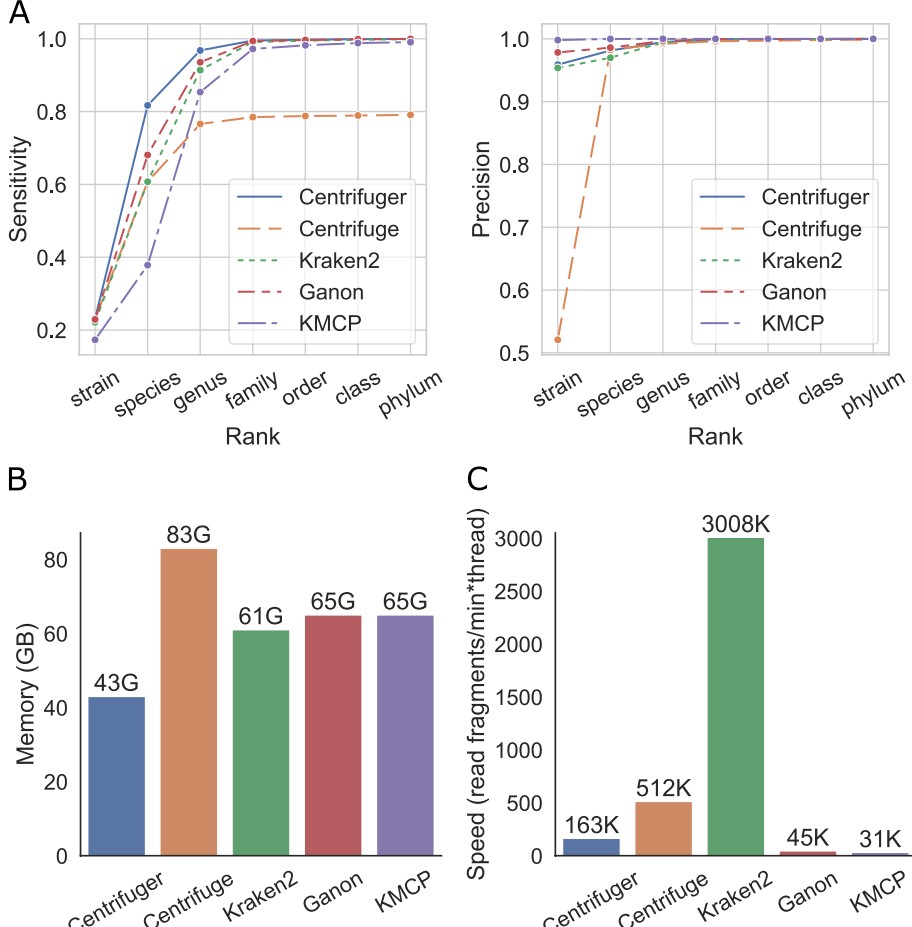

**Fig. 3** Performance of Centrifuger, Centrifuge, Kraken2, Ganon, and KMCP on the simulated data generated from June 2023 RefSeq prokaryotic genomes. **A** Sensitivity (left) and precision (right) of each classifier at various taxonomy ranks. **B** Peak memory usage of each classifier. **C** Classification speed of each classifier with a single thread

We utilized the simulated data set generated by Mason to compare the classifiers' computational efficiency. Centrifuger was the most memory-efficient method, classifying the reads against the 140 billion bp (GBp) database using 43 gigabytes (GB) of memory (Fig. 3B). Methods like Kraken2, Ganon, and KMCP reduce the memory usage by discarding k-mer information. We also explored the space usage of succinct colored k-mer representations [32], which can keep all the k-mer information along with their color (sequence ID) information. We created the index on these RefSeq prokaryotic genomes using Themisto v3.2.1 [33], a pseudoalignment method based on the spectral BWT [34], using a k-mer size of 31. Its index, without the color component, took 44 GB space (the.tdbg file), which was already more than Centrifuger's 41-GB full index size. Since Themisto is not designed for taxonomic classification, we excluded it from other evaluations. Nevertheless, this observation suggests that succinct colored k-mer representations could be memory-efficient enough for read classifications against a large microbial genome database. For the classification speed, Kraken2 was the fastest method. Centrifuger and Centrifuge

were also efficient and processed more than 100,000 read pairs per minute using a single thread (Fig. 3C). Centrifuger was about three times slower than Centrifuge, reflecting the earlier observation that the rank query on RBBWT was three times slower than on an uncompressed data structure. Ganon and KMCP were the slowest methods in the evaluation; they were about 3.6 times and 5.3 times slower than Centrifuger, respectively.

We next evaluated each method's performance on a simulated data set when true genomes were missing in the genome database. We created another index for each method on a trimmed database with 1,931 genomes, where we randomly selected one genome per genus. We then removed all reads originating from the selected genomes in the Mason-generated simulated data. This yielded a simulated data set with about 946K read pairs whose true origin was not in the database. On this trimmed database, Centrifuger, Centrifuge, and Kraken2 had similar accuracy, where they were more sensitive but less precise than Ganon and KMCP (Additional file 1: Fig. S4A). Due to the large discrepancies in sensitivity and precision among the five classifiers, we compared their F1 scores, defined as 2·(sensitivity*precision)/ (sensitivity + precision). Centrifuger, Centrifuge, and Kraken2 achieved very similar F1 scores across the taxonomy ranks. Centrifuger's F1 scores were 4.1–17.1% and 21.6–30.2% higher than Ganon's and KMCP's, respectively, ranging from the genus level to the phylum level (Additional file 1: Fig. S4B). Ganon outperformed other classifiers at the species level, with the F1 score 5.2% higher than Centrifuger's. We observed that the Centrifuger and Centrifuge had almost identical performance on this trimmed database, suggesting that the difference in their performance on the full database was primarily due to the redundancy of the genomes.

In addition to the comparisons on our own simulated data sets, we also evaluated the accuracy of these five classifiers on 10 simulated short-read samples from the Critical Assessment of Metagenome Interpretation 2 (CAMI2) [35] challenge datasets. Each sample has about 6.7 million 150-bp read pairs. Since the truth table from CAMI2 was mostly at the species level, we skipped the strain-level comparison. As in the previous simulated data evaluation, Centrifuger achieved significantly better classification results at species and genus levels than other methods (Fig. 4) and the highest F1 score across all the taxonomy ranks (Additional file 1: Fig. S5). For example, at the species level, the mean sensitivity of Centrifuger was 72.9% and 54.1% higher than Centrifuge's and Kraken2's, and the mean precision was 8.3% and 11.0% higher than Centrifuge's and Kraken2's, respectively. Centrifuger's average sensitivity was 13.7% higher than Ganon's while obtaining almost identical precision at the species level. Though Centrifuger's average precision was 4.8% lower than KMCP's at the species level, its sensitivity was 2.3 times higher than KMCP's. We concatenated the 10 samples into a large data set containing about 67 million read pairs to compare the speed of the classifiers running with eight threads. Kraken2 was the fastest method (finished in about 7 min), followed by Centrifuge (42 min), Centrifuger (1 hour 35 min), and Ganon (3 hours 33 min). With multithreading, Centrifuger was about 2.3 times slower than Centrifuge, reducing the threefold speed difference when running on a single thread. KMCP did not scale well and took more than 18 hours to finish.

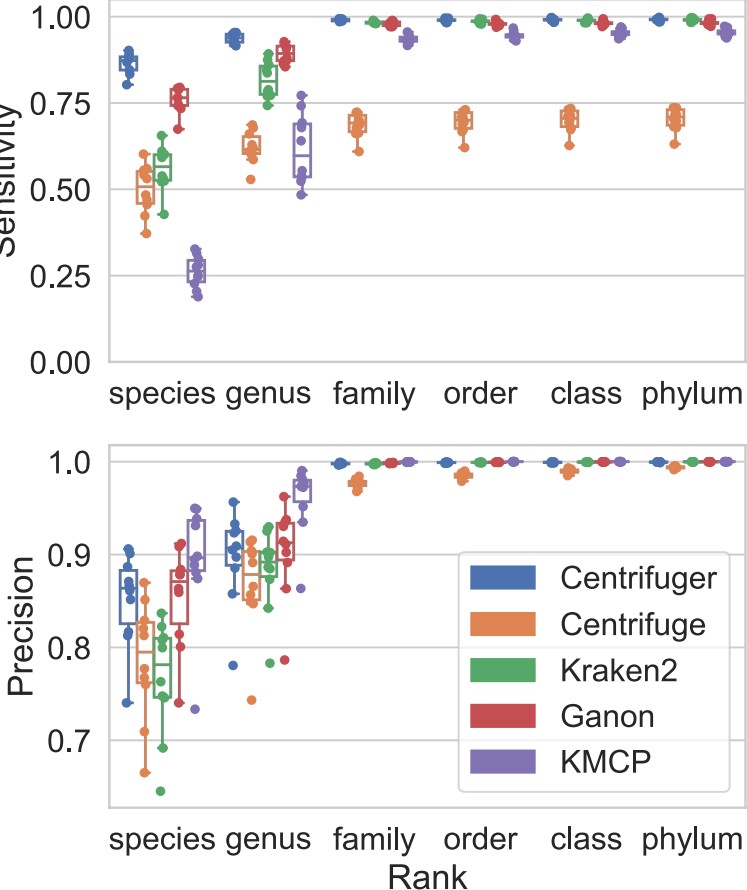

**Fig. 4** Sensitivity (top) and precision (bottom) of Centrifuger, Centrifuge, Kraken2, Ganon, and KMCP at various taxonomy ranks on the 10 simulated data sets from CAMI2

**Performance on classifying bacterial whole-genome sequencing data**

We next evaluated Centrifuger, Centrifuge, Kraken2, Ganon, and KMCP on real bacterial whole-genome sequencing (WGS) data using the same database indexes as the simulated data evaluation. The true taxonomy IDs for each WGS sample were extracted in corresponding SRA RunInfo entries. We considered two scenarios: one where the RefSeq database contained some genomes from the same species (species-in), and one where the database did not include any same-species genomes but did include some same-genus genomes (species-not-in). We collected 100 WGS samples for each scenario, and all the classifiers, except KMCP, successfully processed these samples. KMCP failed to finish SRR23033313 and SRR23885914 in the species-in scenario on our server due to its long running time. Sensitivity and precision were defined in the same way as the simulated data evaluations, and we focused on the accuracy at the species and genus levels for species-in and species-not-in scenarios, respectively. For the species-in scenario, Centrifuger achieved the highest average sensitivity and average precision. In particular, Centrifuger achieved 10.6%, 1.3%, 9.6%, and 101.9% higher average sensitivity, 5.8%, 18.6%, 1.7%, and 6.7% higher average precision than Centrifuge, Kraken2, Ganon, and KMCP, respectively (Fig. 5A). When comparing with KMCP, we excluded the two samples that KMCP did not finish. For the species-not-in scenario, Centrifuger,

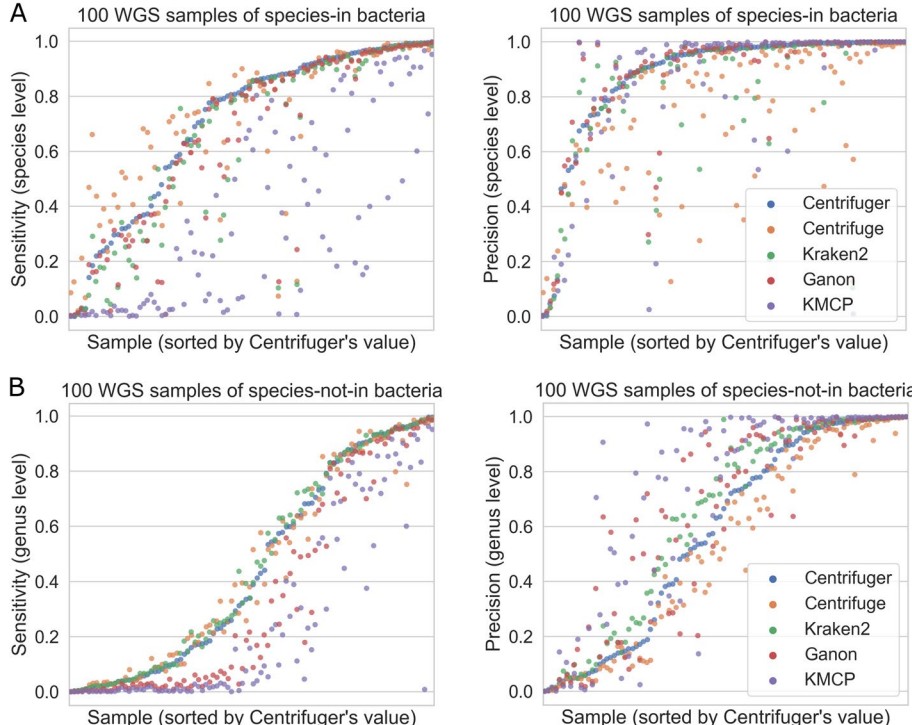

**Fig. 5** Performance of Centrifuger, Centrifuge, Kraken2, Ganon, and KMCP on bacterial WGS data sets. **A** Sensitivity (left) and precision (right) if species of bacteria are present in the database. **B** Sensitivity (left) and precision (right) if species of bacteria are not in the database but their genera are present in the database

Centrifuge, and Kraken2 had comparable sensitivity, Kraken2's precision was 10.2% and 21.2% higher than Centrifuger and Centrifuge, respectively (Fig. 5B). Though Ganon's average precision was similar to Kraken2's and was 11.1% higher than Centrifuger's, its average sensitivity was 22.2% lower than Centrifuger's. KMCP obtained the highest precision, with an average precision 23.1% higher than Centrifuger's, but its average sensitivity was 40.6% lower than Centrifuger's. We further examined the F1 score for each classifier. Kraken2 achieved the highest F1 score, closely followed by Centrifuger, and their F1 scores were consistently greater than Ganon's and KMCP's (Additional file 1: Fig. S6). Our analysis also showed that the species-not-in scenario had inferior accuracy compared with the species-in scenario, suggesting that having a comprehensive database may substantially improve classification results by reducing the species-not-in chance.

**Performance on classifying SARS-CoV-2 Oxford Nanopore WGS data**

When a read can be uniquely classified to a sequence, Centrifuger reports the sequence ID in addition to the taxonomy ID, while many methods like Kraken2 provide only the taxonomy ID information. Centrifuger's additional output is desirable for virus analysis. For example, SARS-CoV-2 variants' genomes are all under the same taxonomy ID 2697049 in RefSeq and GenBank. To explore the effectiveness of sequence-level classification, we downloaded Oxford Nanopore (MinION) WGS data from two SARS-CoV-2 projects with NCBI BioProject accession numbers PRJNA673096 and PRJEB40277, where PRJNA673986 were samples from the USA and PRJEB40227 were samples from

Ireland. For each project, we selected 100 samples with the greatest number of reads (Additional file 1: Table S2). Since RefSeq only had one SARS-CoV-2 sequence, we added the 92 SARS-CoV-2 sequences from GenBank and created indices comprising RefSeq human, prokaryotic, virus, and GenBank SARS-CoV-2 genomes for Centrifuger, Centrifuge, Kraken2, and Ganon, respectively. We also incorporate two long-read taxonomic classifiers: MetaMaps [36] and Taxor [37]. However, due to the large memory requirement for running MetaMaps and a different taxonomy structure when building the Taxor's index, we tested these two methods with indices that only contained the 93 SARS-CoV-2 genomes. We added the parameter "--rel-cutoff 0.12 --rel-filter 0.9" to Ganon for long reads as mentioned by the experiments in Taxor. Centrifuger classified about 99.95% of all the input reads to the taxonomy ID 2697049 on average, while Kraken2 and Ganon were slightly less sensitive and classified 99.90% and 98.96% reads to the ID 2697049 on average, respectively. Centrifuge had a slightly different LCA search implementation, so it classified 98.90% of the reads to either ID 2697049 or ID 694009, where ID 694009 was the parent of ID 2697049 on the taxonomy tree. Despite using a SARS-CoV-2-only database, MetaMaps and Taxor only classified 78.04% and 70.91% reads on average, respectively. The lower sensitivity for MetaMaps was mainly due to its default setting of skipping reads shorter than 1000 bp, while it mapped almost all the remaining reads of sufficient length. When looking at the sequence-level classification, Centrifuger assigned 23.7% of the input reads with unique sequence IDs across all the samples. For the other tested methods, only Centrifuge, Ganon and Taxor reported sequence-level classifications, but they uniquely classified 15.1%, less than 0.1% and about 0.1% of the reads, respectively.

The large number of sequence-level classifications from Centrifuger allowed us to observe that the read fraction for each variant, namely sequence ID, in the Irish samples and US samples showed distinct patterns (Fig. 6A, heatmap with raw read fraction in Additional file 1: Fig. S7A). This observation was also supported by Centrifuge's sequence-level classification results (Additional file 1: Fig. S7B). We further conducted a principal component analysis (PCA) based on each sequence's read fraction, normalized by the number of reads with sequence IDs. Samples from the US and Ireland were well separated into two clusters (Fig. 6B) based on the first two principal components (PCs), suggesting that variants found in the two projects may have different sequence features. When inspecting the SAR-CoV-2 variant that contributed the most to the PC1 relative to the contribution to PC2, we found that variants detected in the Irish samples might have homologous regions to MT019531.1 while US samples did not (Fig. 6C). On the other hand, when checking PC2's major contributors, the MT159706.2 variant was commonly detected in both projects, suggesting that PC1 captured the project-specific variants or batch effects.

## Discussion

We conducted comprehensive benchmarks to demonstrate that the RBBWT can significantly reduce the memory usage of the FM-index built over a microbial genome database. For additional space savings, we store the sequence ID rather than the full coordinate information in the sampled suffix array, a strategy also used in Centrifuge. The space for the sampled sequence IDs is further trimmed by bit-compact

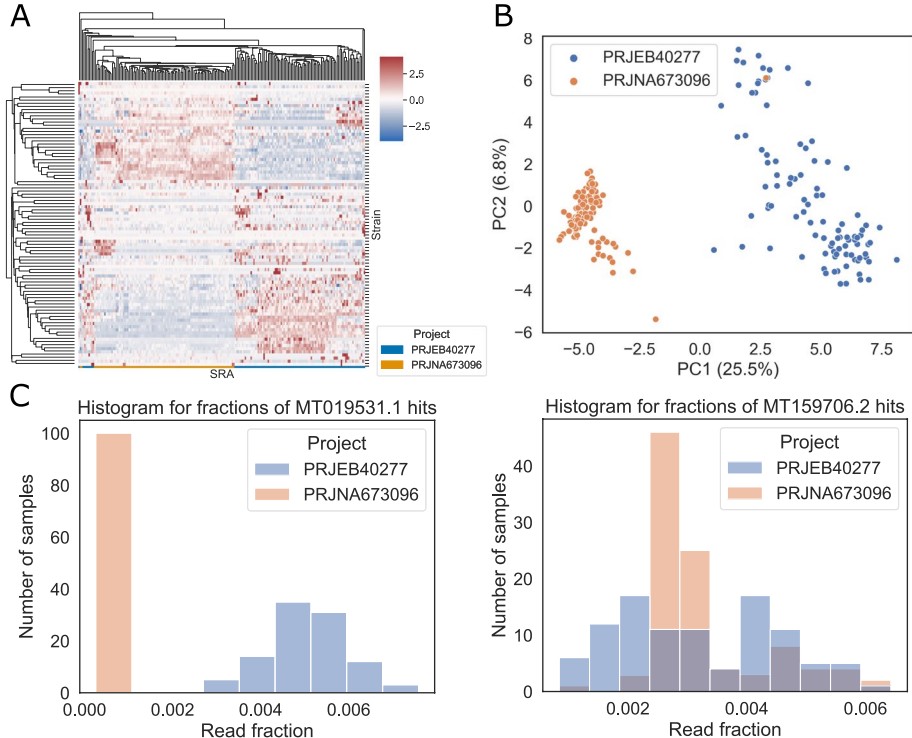

**Fig. 6** Sequence-level classification for SARS-CoV-2 WGS samples with Centrifuger. **A** Fractions of reads hit on each SARS-CoV-2 variant. The rows are SARS-CoV-2 variants present in the RefSeq and GenBank, and the columns are the Oxford Nanopore WGS samples. The value for each row is standardized as *z*-scores. **B** PCA based on the read fraction for each variant. The numbers in the parenthesis are the variance fraction explained by each PC. **C** Left: histogram of read fractions classified to MT019531.1 which has the most significant contribution to PC1 relative to its PC2's contribution; right: histogram of read fractions classified to MT159706.2 which has the most significant contribution to PC2 relative to its PC1's contribution

representation, making it a less impactful factor in space usage. For example, in Centrifuger's 41-GB index of RefSeq prokaryotic genomes, 23 GB was for the RBBWT and 17 GB was for the sampled sequence IDs. Nevertheless, the overall space complexity of Centrifuger is still O($n$) words due to the structure of sampled sequence IDs, making it worse than r-index's O($r$) words in the future as the repetitiveness of the genome database may grow fast. Though users can select a sparser sampling rate to maintain the space usage, this is at the expense of time efficiency. For instance, the index size became 32 GB when increasing the sampling rate from the default 16 to 32, but the classification speed of a thread decreased from 163K read/min to 102K read/min. Future work is needed to design a representation of the sampled sequence ID in sublinear space without sacrificing classification speed for microbial genomes.

Resolving the sequence IDs for each match is a time-consuming step in Centrifuger, especially when a match hits many sequences. Centrifuger's current implementation follows the traditional FM-index paradigm, by repeatedly applying the LF mapping for each hit until reaching a sampled sequence ID. Alternative techniques like the document array profile [38] support rapid sequence ID retrieval, but are designed for highly repetitive genomes like human pangenome. Therefore, a memory-efficient algorithm for sequence ID resolving in microbial genomes is still needed. Since the

default setting in Centrifuger is to report the LCA taxonomy ID for a read, algorithms like KATKA [39] that directly find the LCA taxonomy ID for a k-mer might suggest ways to avoid the overhead of repeated LF mappings in Centrifuger as well.

The current Centrifuger index stores nucleotide-based sequences. The Kraken2 and Kaiju [40] studies, however, observed that translated search, i.e., finding matches based on amino acids by translating nucleotides, could improve the classification accuracy for viral genomes. Since the wavelet tree data structure supports arbitrary alphabet sets, the RBBWT representation can be naturally extended to process amino acid sequences. RLBWT and RBBWT's implementations are scalable for large alphabet set sizes, with a factor of $\log(\sigma)$ or the entropy in the space complexity. We found an alternative form of run-length encoded BWT, exported from ropeBWT2 [41] (with -dRo option) as the Fermi's [42] format. RopeBWT2's representation was 12.5% smaller than RLBWT on average with comparable rank query speed as RLBWT when compressing *Escherichia fergusonii*'s genomes. The slimmer size of ropeBWT2's output might be attributed to its implementation being designed for small $\sigma$, where its space complexity is linear to $\sigma$. This may suggest that the efficiency of RBBWT and Centrifuger could be further improved for nucleotide search with tailored implementations that are less scalable for $\sigma$.

Besides taxonomic classification, another important problem in metagenomic data analysis is taxonomic profiling, i.e., determining the abundance for each species or at user-specified taxonomic ranks. It is feasible to profile the abundances directly without taxonomic classification for reads, such as in Meta-Kallisto [43] and Sylph [44]. However, many profiling methods still require taxonomic classification results which can identify species of low abundance with few reads supporting them. For instance, Bracken uses Kraken2's input for taxonomic profiling with a Bayesian method [45]. Ganon, KMCP, and Taxor, which are benchmarked in this work, need to conduct taxonomic classifications before profiling. Centrifuge integrates the abundance estimation based on the Expectation–Maximization (EM) algorithm [46] internally. After the release of Centrifuge, some methods, like AGAMEMNON [47] and Centrifuge+[48], improve the profiling accuracy by adjusting the likelihood function and the EM algorithm procedure. We still need to systematically compare these profiling techniques and either integrate them into Centrifuger or make Centrifuger's output compatible with these methods in the future. For example, Centrifuger provides the script to summarize the classification results into a Kraken-style report file that can serve as the input for Bracken.

## Conclusions

Centrifuger is an efficient and accurate taxonomic classification method for processing sequencing data, including metagenomic sequencing data. Centrifuger adopts a novel compact data structure, run-block compressed sequence, to achieve sublinear storage space for BWT sequence without sacrificing much time efficiency. Specifically, Centrifuger can represent the 140 GBp Refseq prokaryotic genomes with an index of size 41 GB and classifies about 163K microbial reads every minute per thread. Furthermore, the lossless representation nature and the unconstrained pattern match length help Centrifuger achieve significantly better accuracy, in both sensitivity and precision, for classifications at the species or genus level. We expect that Centrifuger will contribute to

microbiome studies by allowing the incorporation of the recent, more comprehensive microbial genome database. Centrifuger is a free open-source software released under the MIT license and is available at https://github.com/mourisl/centrifuger.

## Methods

### Sequencing data and benchmark details

We generated the simulated data from the current RefSeq database using Mason v0.1.2 [30] with the option "illumina -pi 0 -pd 0 -pmms 2.5 -s 17 -N 2000000 -n 100 -mp -sq -hs 0 -hi 0 -i", which simulated two million 100-bp read pairs with 1% error rate. We further filtered the reads from the sequences without taxonomy information and kept the first one million read pairs as the final simulated data set. We also obtained one million randomly selected 100-bp read pairs using the simulator ART v2.5.8 [31] with Illumina profile setting ("art_illumina -ss HS25 -m 1000 -s 100 -l 100 -f 0.003") followed by the same filtration procedure as was used for Mason. The error rate of Art-generated simulated data was estimated to be around 0.15% by examining the aln file produced by Art. In addition to our own simulated data, we downloaded the 10 simulated samples from the CAMI2 Challenge datasets at https://frl.publisso.de/data/frl:6425521/strain/short_read/ by lexicographical order, i.e. sample_0 to sample_9. For the bacteria WGS data, we first downloaded the RunInfo from NIH NCBI SRA using the search word "("Bacteria"[Organism] OR "Bacteria Latreille et al. 1825"[Organism]) AND ("2022/01/01"[MDAT]: "2023/08/01"[MDAT]) AND ("biomol dna"[Properties] AND "strategy wgs"[Properties] AND "platform illumina"[Properties] AND "filetype fastq"[Properties])". Then based on whether the species or genus is present in the RefSeq database, we randomly pick 100 SRA IDs (Additional file 1: Table S3), without repeating species or genus, for species-in and species-not-in evaluations, respectively.

We benchmarked the performance of Centrifuger v1.0.1, Centrifuge v1.0.4, and Kraken2 v2.1.3, Ganon v2.0.0, and KMCP 0.9.4 in this study. For the application on long-read data sets, we also tested MetaMap with GitHub commit ID 633d2e0 and Taxor v0.1.0. Taxonomy information and microbial genomes were downloaded using the "centrifuger-download" script in June 2023. Each classifier was used to build its own index on dustmasked [49] genome sequences. Kraken2 used its own built-in masking module. Though we used Ganon v2.0.0 in evaluations, we failed to create the index with the hierarchical interleaved bloom filter data structure implemented in this version [50]. Therefore, we used the index based on the interleaved bloom filter (--filter-type ibf) for Ganon, and we referred to this method as "Ganon" rather than Ganon2. The commands used to run each method are listed in Additional file 1: Table S4. Methods like Centrifuge, Ganon, and KMCP might report multiple equally good taxonomy IDs for a read, and we merged them into LCAs before the evaluations. Ganon's default option for coalescing the taxonomy IDs of a multiple-classified read is to reassign the read to the taxonomy ID with the highest abundance inferred by the EM algorithm using the initial classifications. This approach substantially improves Ganon's classification sensitivity at lower taxonomy levels, leading to a higher F1 score. For example, in the Mason-generated simulated data, Ganon with reassignment's F1 score at the species level was 1.3% and 12.1% higher than Centrifuger's and Ganon using LCA's, respectively (Additional file 1: Fig. S8). We implemented a similar workflow to reassign the multiple-classified reads

from Centrifuger. Specifically, we ran Centrifuger with "-k 5" so that the initial classification for a read could include up to five equally good classification results. We then calculated the abundance for each taxonomy ID in the taxonomy tree based on the number of reads classified to this taxonomy node and its subtree. For a multiple-classified read, we included its count starting from its LCA. Lastly, we reassigned the taxonomy ID with the highest abundance among the initial results to a read as the final classification result. When there were multiple highest-abundance taxonomy IDs for a read, we took their LCA as the final result. We observed this reassignment strategy without the EM algorithm improved Centrifuger's classification results at lower taxonomy levels too, with the F1 score 2.7% higher than Ganon's reassignment results at the species level (Fig. S8). However, the strategy of hard reassignment based on the taxonomic profiling result may result in systematic underestimations of taxa with lower abundances. Therefore, we continue to utilize the LCA strategy to process multiple classified reads, where the results can be directly used for downstream analyses, including taxonomic profiling.

All the benchmarks were conducted on the 2.8 GHz AMD EPYC 7543 32-core processor machine with 512 GB memory. The memory footprint was measured as the "Maximum resident set size" value from the "/usr/bin/time -v" command. When measuring speed, each classifier was run four times. The reported classification speed was calculated by taking the fastest runtime after excluding index loading time.

### Run-block compressed sequence

Run-block compressed sequence is a compact data structure supporting rank queries for any position in a sequence. For the input sequence $T$ of length $n$ and alphabet set $\Sigma$ of size $\sigma$, we first partition $T$ into equal-size substrings (blocks), $T_1, T_2, \ldots, T_m$, where $m = \lceil \frac{n}{b} \rceil$ and $b$ is the block size. The first component of the run-block compressed sequence is a bit vector $B_R$ of size $m$ indicating whether the corresponding block is a run block, i.e., a block consisting of one alphabet character repeated $b$ times. We will then split $T$ into two substrings, by concatenating run blocks and non-run blocks, i.e., $T_{R'} = T_{i_1} T_{i_2} \ldots T_{i_l}, T_P = T_{j_1} T_{j_2} \ldots T_{j_{m-l}}$, and $T_{i_k}$ is the $k$-th run blocks in T with the alphabet $\sigma_{i_k}$. In the notation, we will use the subscript "R" to denote run-block compressed sequence, and "P" to represent the plain uncompressed sequence. Since the last block can still be determined as a run or non-run block even if it is shorter than $b$, we can assume that $n$ is divisible by $b$ for simplicity. $T_{R'}$ can be losslessly represented as $T_R = \sigma_{i_1} \sigma_{i_2} \ldots \sigma_{i_l}$, where $|T_R| = \frac{|T_{R'}|}{b}$. The space saving comes from using one character to represent a run block of size $b$, a strategy we call run-block compression. For example, for a sequence $T=$"AAAAACGTAAAA", when $b=4$, it will be split into "AAAAAAAA" and "ACGT" guided by $B_R = 101$. For the subsequence formed by the run blocks, we will use one character to represent each block in it. Therefore, the example sequence $T$ will be represented by two sequences $T_R=$"AA" and $T_P=$"ACGT", reducing the original length from 12 to 6 characters (Fig. 1). We next show that run-block compression allows fast rank queries and sublinear space usage as the repetitiveness in the sequence increases. The rank query is the core operation in LF mapping during the backward search in FM-index.

**Theorem 1** *The time complexity for rank query on run-block compressed sequence is* $O(log\sigma)$.

*Proof*: We will use the function $rank_c(i, T)$ to denote the rank for the alphabet $c$ at position $i$ of text $T$, where the index is 1-based. Equivalently, $rank_c(i, T)$ counts the number of $c$'s that occur before $T[i]$, including $T[i]$. We can decompose the $rank_c(i, T)$ to the sum of corresponding ranks with respect to $T_R$ and $T_P$. There are two cases, depending on whether $i$ is in a run block or not. Let $k$ denote the block containing $i$, namely $k = \lceil \frac{i}{b} \rceil$. We compute the number of run blocks and non-run blocks before the block containing $k$ as $r_r = rank_1(k, B_R)$, and $\overline{r_r} = rank_0(k, B_R) = k - r_r$, respectively. The $rank_1(k, B_R)$ is the conventional rank query on bit vectors counting the number of 1 s before or on position $k$ in $B_R$. With these notations, we can write the equations to compute $rank_c(i, T)$.

When $i$ is in a run block, i.e., $B_R[k] = 1$, we have:

$$rank_c(i, T) = b(rank_c(r_r, T_R)) + rank_c(\overline{r_r}b, T_P) + I_{T_R[r_r]=c}((i-1)\%b + 1 - b)$$

Where $I_{T_R[r_r]=c}$ is an indicator that equals 1 if the subscript is true and equals to 0 otherwise. The last term is the special treatment if $i$ is in a run block with alphabet character $c$. When $i$ is in a non-run block, we have:

$$rank_c(i, T) = b(rank_c(r_r, T_R)) + rank_c((\overline{r_r} - 1)b + (i-1)\%b + 1, T_P)$$

If we apply the wavelet tree to represent $T_R$ and $T_P$, then $rank_c(i, T)$ can be answered by at most two wavelet tree rank queries, one rank query on bit vector $B_r$, and one wavelet tree access on $T_R$. Therefore, the total time complexity is $3O(log\sigma) + O(1) = O(log\sigma)$.

The naïve implementation for calculating $I_{T_R[r_r]=c}$ is to access the value of $T_R[r_r]$, requiring $O(log\sigma)$ time if using wavelet tree. We note that the $I_{T_R[r_r]=c}$ can also be inferred during the wavelet tree's rank query on $T_R$, by checking whether the bits labeling the relevant root-to-leaf path form the bit representation for c. This strategy further accelerates the $rank_c(i, T)$ operation, and is also applicable to other compressed sequence representations, including RLBWT.

**Theorem 2** *The space complexity of run-block compressed sequence is* $O(\frac{n}{\sqrt{l}}log\sigma)$ *bits, where* $l = \frac{n}{r}$ *is the average run length and r is the number of runs in the sequence.*

*Proof*: The key observation is that each non-run block contains at least one run head. Therefore, we have at most $r$ non-run blocks. As a result, the minimum length of $T_R$ is $\frac{n}{b} - r$, and the maximum length of $T_P$ is $rb$. The length of the $B_R$ is $\frac{n}{b}$.

We can use wavelet trees to represent the run-block subsequence and the plain subsequence. Let $A$ be the number of bits to represent one character in the wavelet tree, then the asymptotic total space usage in bits is $\frac{n}{b} + A\left(\frac{n}{b} - r\right) + Arb$, where the terms are for $B_R$, $T_R$, and $T_P$, respectively. We can rewrite the space usage bound as:

$$S(b) = Arb + \frac{1 + An}{b} - Ar,$$

which is minimized when $b = \sqrt{\frac{1+An}{Ar}}$. Because $b$ is an integer, we take the block size as $b^* = \lceil \sqrt{\frac{1+An}{Ar}} \rceil$. Substituting this for $b$, we have:

$$S(b^*) = Ar\lceil\sqrt{\frac{1+An}{Ar}}\rceil + \frac{1+An}{\lceil\sqrt{\frac{1+An}{Ar}}\rceil} - Ar < \sqrt{Ar(1+An)} + \sqrt{Ar(1+An)} = O(A\sqrt{nr}),$$

where the inequality is based on $\sqrt{\frac{1+An}{Ar}} \leq \lceil\sqrt{\frac{1+An}{Ar}}\rceil < \sqrt{\frac{1+An}{Ar}} + 1$. We can rewrite $S(b^*)$ by using the definition of $r = \frac{n}{l}$, obtaining $S(b^*) = O\left(A\frac{n}{\sqrt{l}}\right)$. To ensure the worst rank query time on the data structure is small, the wavelet tree is in the shape of a balanced binary tree in our implementation, and $A = O(\log\sigma)$. Further space could be reduced if we use techniques like Huffman-shaped wavelet tree [51], and $A$ will be $O(H_0(T_R))$ and $O(H_0(T_P))$ for $T_R$ and $T_P$, respectively, where $H_0$ is the Shannon entropy of the sequence.

The $b^*$ found in the proof is to bound the worst-case space usage, where each non-run block has exactly one run head in the middle. The optimal block size can be different. For example, when every run has an identical length, the optimal block size is $\frac{n}{r}$ and every block is run-block compressible, yielding $O\left(\frac{n}{l}\log\sigma\right)$-bit space complexity. To find the appropriate block size efficiently, we search the size of powers of 2, e.g., 4, 8, 16,.., and select the block size $\widehat{b}$ with the least space usage among them. Suppose $\widehat{b}'$ is the smallest power of 2 that is larger or equal to the block size $b^*$ defined in the proof, then we have $b^* \leq \widehat{b}' \leq 2b^*$. Therefore, $S\left(\widehat{b}\right) \leq S\left(\widehat{b}'\right) \leq 2Arb^* + \frac{1+An}{b^*} - Ar \leq 2S(b^*) + Ar = O\left(\frac{An}{\sqrt{l}}\right)$, where the second inequality is by applying $b^* \leq \widehat{b}' \leq 2b^*$. Therefore, the block size inferred from inspecting powers of 2 is not a bad estimator and gives the same asymptotic space usage as $b^*$ in the worst case. Furthermore, since $Ar \leq A\sqrt{nr} < S(b^*)$, $S\left(\widehat{b}\right)$ is no more than three times of the $S(b^*)$. To reduce the bias of the sparse search space, we also inspect the space usage of block sizes $b^*$ and $\frac{3\widehat{b}}{2}$ before making a final decision. When $l$ is small, the block size minimizing the overall space usage is the length of the genome ($T_P = T$), and run-block compression is equivalent to the wavelet tree representation. In practice, Centrifuger uses the first one million characters of the BWT sequence instead of the full sequence to infer the block size.

### Index construction

Centrifuger uses the blockwise suffix sorting algorithm [52] to build its index, as in Bowtie [53] and Centrifuge. The advantage of blockwise suffix sorting is to control the overall memory footprint and parallelize the construction procedure. The array that holds the BWT sequence is pre-allocated, and the sequence is filled in block-by-block as blocks of the suffix array are constructed. Another important component in the FM-index is the sampled suffix information. During construction, the index stores the genome coordinate information for every 16th offset on the BWT, which can be adjusted by the user. After that, the offsets are transformed into sequence IDs and using a bit-efficient representation of the IDs. For example, the RefSeq prokaryotic genome database contained 75,865 sequences (including plasmids) from 34,190 strains with complete genomes, so sequence IDs can be distinguished by a 17-bit integer. Instead of saving the IDs in an array of 32-bit integers, the bit patterns are stored consecutively without any wasted space. Therefore, the total size for the bit-compact array for the sampled ID list is $17 \cdot m$ bits, or $0.26 \cdot m$ 64-bit words, where $m$ is the number of sampled IDs.

**Taxonomic classification**

For taxonomic classification, Centrifuger follows Centrifuge's paradigm by greedily searching for semi-maximal exact matches, where only one side of a match cannot be extended to form a longer match to the database (Fig. 1). For each read pair, Centrifuger searches the matches twice, one using the forward strand and the other using the reverse-complement strand. Using the forward strand search as an example, Centrifuger starts from the last position of the read and extends the match backward as much as possible using LF mapping until the match cannot hit any sequence in the index. In the example in Fig. 1, the first match is the right-most 60 bp. Centrifuger will then skip the next base on the read, as a putative variant or sequencing error, and start a new search. As a result, the next search starts from the 62nd bp, counting from the right end, and finds a match of length 39.

To find the best taxonomy ID for the read, Centrifuger scores the sequence IDs retrieved from the matches. Let $l_M$ denote the length of a match $M$. Centrifuger will filter a short match $M$ if $2n/4^{l_M} > 0.01$, as it is likely a random match. For example, we kept the matches with lengths greater or equal to 23 as valid matches when classifying reads using the 140 GBp RefSeq prokaryotic database. Each valid match $M$ of length $l_M$ will contribute a score $(l_M - 15)^2$ to the corresponding strand, and the matches from the less-scored strand will be removed. Centrifuger resolves sequence IDs contained in each valid match by using LF-mapping in the compressed FM-index to find sampled sequence IDs. Specifically, each match $M$ corresponds to an interval on the BWT sequence, denoted as $[s_M, e_M]$, then we will resolve the sequence ID for each position on the BWT sequence between $[s_M, e_M]$, by applying LF-mapping. The LF-mapping procedure for resolving the position $p \in [s_M, e_M]$, terminates when it reaches a position $p'$ on the BWT sequence with a sampled sequence ID, i.e., $p' \bmod 16 = 0$ with the default parameter. The sampled sequence ID at $p'$ is the sequence ID corresponding to $p$. In the example of Fig. 1, the Centrifuger found the first 60-bp exact match $M_1$ hit sequences with IDs X, Y, and Z, and the second 39-bp exact match $M_2$ were from sequence IDs W and X. If both strands have an equal score, the sequence ID will be resolved for both strands. For each resolved sequence ID, Centrifuger will sum up its scores across the matches using the formula $\text{score}(\text{sequence}I) = \sum_{M \in \text{sqeuence}I}(l_M - 15)^2$, where $M \in \text{sequence}I$ means the match $M$ is found in the sequence with ID $I$. In the example of Fig. 1, the score for sequence ID X is $(60 - 15)^2 + (39 - 15)^2 = 2601$ as both matches hit this sequence. This scoring function was empirical and was designed during the development of Centrifuge. The sequence IDs with the highest score and their corresponding taxonomy IDs will be reported as the final classification result for the read, where the example read in Fig. 1 is classified to the sequence X. When the number of highest-scoring sequence IDs is more than the report threshold that can be specified by the user (default 1), Centrifuger will merge the IDs to their LCAs in the taxonomy tree until the number is within the threshold.

For a match that hits many sequences in the database, i.e., $e_M - s_M + 1$ is large, Centrifuger will resolve the sequence IDs for at most 40·report_threshold entries evenly distributed in the BWT interval. For example, if $s_M = 100$ and $e_M = 490$, then Centirufger will resolve the 40 sequence IDs for positions at 100, 110, 120,.., 480, and 490 on the BWT sequence with the default setting. Though this is a heuristic that can

cause the algorithm to miss the true genome of origin, it is likely to generate scores for sequences in the same phylogeny branch and may help identify the correct taxonomy IDs at higher levels. This is the main difference between Centrifuge and Centrifuger in the classification stage, where Centrifuge ignores a match $M$ if $e_M - s_M + 1 > 200$ with the default setting.

### Hybrid run-length compression

In addition to the run-block compression, we designed another compression scheme called hybrid run-length compression, using run-length compression for each fixed-size block. Hybrid run-length compressed BWT uses the same amount of space as the wavelet tree representation when the repetitiveness of the sequence is low, and its space usage converges to the RLBWTs when the repetitiveness grows. In our implementation, a block is marked as run-length compressible ($B_R = 1$) if its average run-length is more than six based on the comparison between RLBWT and wavelet tree (Fig. 2A, B). The substrings from blocks are separated into two subsequences based on $B_R$, and the subsequences will be concatenated into two sequences, $T_R$ and $T_P$, respectively. $T_R$ will be compressed by the run-length method as in RLBWT, and $T_P$ will be represented as a wavelet tree. The block size is inferred in the same fashion as in RBBWT. The rank query on the hybrid run-length compressed sequence is like the run-block compression, where we combine the ranks from $T_R$ and $T_P$, making it slower than the rank query on RLBWT. The idea of hybrid run-length compression is similar to the wavelet tree when using fixed-block boosting [54]. Our implementation avoids explicitly recording the accumulated count at the beginning of a block and is therefore well suited to mildly repetitive sequences needing small blocks. For example, the block size of the hybrid run-length compressed BWT was only 12 for the genus *Legionella*'s genomes. Despite being less computationally efficient than other representations, the hybrid run-length compressed BWT is flexible and allows methods to handle various texts with a wide range of repetitiveness without altering the underlying data structure.

### Supplementary Information

**Additional file 1**: Table S1. The classification accuracy at various taxonomy ranks in the Mason-generated simulated data Table S2. SRA IDs of the samples used in the SARS-CoV-2 sequence-level classification analysis. Table S3. SRA IDs of the samples used in the bacterial WGS classification evaluations. Table S4. Running commands for the classifiers used in the evaluations. Fig. S1. Space usage of the wavelet tree, RLBWT, hybrid run-length compression and RBBWT when adding genomes with the species *Escherichia fergusonii* (taxonomy ID 564) and the genus *Legionella* (taxonomy ID 445) Fig. S2. Space usage of the wavelet tree, RLBWT, hybrid run-length compression and RBBWT when adding genomes with species *Chalmydia trachomatis* (taxonomy ID 813) and the genus *Chalmydia* (taxonomy ID 810). Fig. S3. Sensitivity and precision of Centrifuger, Centrifuge, Kraken2, Ganon, and KMCP on the simulated data generated from June 2023 RefSeq prokaryotic genomes using ART. Fig. S4. Performance of Centrifuger, Centrifuge, Kraken2, Ganon, and KMCP on the simulated data when classifying against a trimmed database that has one genome per genus and does not contain the true origins of the reads. Fig. S5. F1 scores of Centrifuger, Centrifuge, Kraken2, Ganon, and KMCP at various taxonomy ranks on the 10 simulated data sets from CAMI2. Fig. S6. F1 scores of Centrifuger, Centrifuge, Kraken2, Ganon and KMCP on bacterial WGS data sets, where the species of the bacteria are not in the database but their genera are present in the database. Fig. S7. Cluster heatmaps for read fractions in the SARS-CoV-2 sequence-level analysis. Fig. S8. Sensitivity, precision and F1 score of Centrifuger and Ganon using LCA and reassignment based on taxa quantification at various taxonomy levels.

**Additional file 2**. Review history.

### Acknowledgements
We thank Dr. Daehwan Kim and Dr. Florian Breitwieser for the foundation work in Centrifuge. We also thank Dr. Heng Li and Dr. Shannon Soucy for many helpful discussions.

### Peer review information
Feng Gao and Kevin Pang were the primary editors of this article and managed its editorial process and peer review in collaboration with the rest of the editorial team.

### Review history
The review history is available as Additional file 2.

### Authors' contributions
 L.S. conceived the project. L.S. and B.L. designed the algorithm. L.S. implemented and evaluated the software. L.S. and B.L. wrote the manuscript.

### Funding
This work is supported by the NIH grants P20GM130454 (Dartmouth), 3P20GM130454-05WS (Dartmouth), R01HG011392 (B.L.), and R35GM139602 (B.L.).

### Availability of data and materials
The source code of Centrifuger is available at https://github.com/mourisl/centrifuger [55]. The source code of the version Centrifuger v1.0.1 used in this study is also available at Zenodo https://doi.org/10.5281/zenodo.10938378 [56]. The code for the evaluations and experiments is available at https://github.com/mourisl/centrifuger_evaluations [57]. The NCBI SRA accession numbers of the WGS data sets are listed in Additional file 1: Table S2 and Table S3. The index for the RefSeq prokaryotic, human, virus, and GenBank SARS-CoV-2 variants is available at Zenodo https://doi.org/10.5281/zenodo.10023239.

## Declarations

### Ethics approval and consent to participate
Not applicable.

### Competing interests
The authors declare that they have no competing interests.

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

## 