## [**Additional**
**file 2**. Review history. · Genome Biology]

Review History

First round of review

Reviewer 1

Were you able to assess all statistics in the manuscript, including the appropriateness of statistical tests used? There are no statistics in the manuscript.

Were you able to directly test the methods? No.

Comments to author:

The authors describe an extension of the already published tool Centrifuge. One main contribution of the work is the definition of a clever, and quite practical compression of the standard FM index used in Centrifuge, named the RunBlock compressed BWT. While the idea is quite standard, it was to my knowledge never published and makes this paper alone already worth publishing. The RBBWT simply divides the BWT into blocks of fixed length. Then the BWT is divided into two parts, the concatenation of blocks that have more than one character and the concatenation of blocks that consists of a single character, where per block only the one character is stored. Access to the BWT (or more important rank queries) then be emulated by three rank queries. One for the indicator bit vector and one each for the two strings.

This is a more practical solution than the RL encoded BWT, that is in theory optimal, but incurs quite a high run time penalty for rank queries and hence for exact search.. The authors then compare the run time

and space requirements of their implementation to the standard BWT, the RL compressed BWT and a hybrid solution.

As expected their run time is only about 3 times worse than the uncompressed version.

They also show that the resulting tool Centrifuger does perform similarly to Centrifuge and Kraken2. While it is not a full comparison (also to other tools like clark or ganon), it is sufficient to prove the claimed improvements of the RBBWT.

I have a number of minor comments that might improve the manuscript.

1) Please give more details of how the Sequence ID sampling is done. I assume that you mark text positions? If so, you would need to mark the beginning of an ID run in addition to the sampling rate markings. Please give more details how it is done.

2) it is usually a "sampled suffix array" not "subsampling"

3) it is "substrings" not "substrings"

4) The performance in classification (sensitivity) differs between centrifuger and centrifuge. Why is that if you use the same search paradigm as in centrifuge? Can you elaborate more on the differences?

Reviewer 2

Were you able to assess all statistics in the manuscript, including the appropriateness of statistical tests used? There are no statistics in the manuscript.

Were you able to directly test the methods? No.

Comments to author:

The paper further develops a metagenomics classification tool Centrifuge to become faster, more space-efficient, and accurate. This is done by replacing the FM-index of Centrifuge with an index that exploits run-length compressibility of the Burrows-Wheeler transform (BWT). Instead of taking an off-the-shelf tool like r-index for this purpose, the authors develop a new fixed-block scheme to partition BWT into run-blocks and non-run-blocks. They show that the optimal block size can be computed efficiently. Furthermore, the fixed-block scheme enables simpler computation of rank-queries to enable faster backwards search.

Authors first compare their approach experimentally to other FM-index implementations of which some exploit run-length compression. The space and time of the new index are competitive.

Then the authors move on to proposing some heuristic changes to Centrifuge and show that these changes improve its accuracy. Several experiments are conducted on variety of dataset to show improved accuracy against k-mer based indexes. However, the approach is still much slower than k-mer approaches limiting somewhat the scalability.

The compression approach is very simple and easy to follow. However, authors refer to Figure 1 on many occasions, but that figure is very condensed and does not explain much. Important concepts like how the scores are defined for the matches are defined in confusing way. Basically, many concepts related to how Centrifuge works should be explained better.

Specific comments:

Background: You say "We previously developed... Centrifuge". However, [18] has many authors not in this submission, so I find "we" not suitable here. Maybe rephrasing "The first author co-developed...".

Page 9, line 4: "in bit is"  "in bits is"

Page 14, line 13, Taxonomic classification: Please define what semi-maximal match means and explain how you find them using backward search. You refer to Figure 1, but it does not help in figuring out what is happening here.

... line 19: "Each valid match"  "Each valid match of length ℓ ".

.. lines 24-26: The definitions do not seem to make any sense. What do you mean by " i in m means match m hits.."? There are many undefined terms here, like ℓ_m and notation i in m .

Reviewer 3

Were you able to assess all statistics in the manuscript, including the appropriateness of statistical tests used? Yes, and I have assessed the statistics in my report.

Were you able to directly test the methods? Yes.

Comments to author:

Song and Langmead presents Centrifuger, a new microbial sequence classification program, defined as an improvement of Centrifuge, a previous method published in 2016.

The manuscript is well written, however I'd like to bring some issues to the authors' attention:

Minor issues:

- In Background, end of 2nd paragraph: definition of n for $O(n)$ is missing.
- In Background, last sentence of the 3rd paragraph: "Inspired by the observation" -> "Inspired by this observation".
- In Results, authors wrote "we downloaded the first 10 simulated samples from the CAMI2 Challenge datasets at https://frl.publisso.de/data/frl:6425521/strain/short_read/". But what is specifically the order used to select these first 10 (Based on file size? date modified? lexicographic? etc.)? Please list the IDs/sample names so we can reproduce your results.

Major issues:

In Methods:

- Why do the authors compare Centrifuger against only 2 other methods? I understand these 2 methods come from the same lab/university thus I'd recommend to include methods from other research groups to present a comprehensive comparison. Methods like MetaPhlan2, CLARK, Ganon and KMCP that are cited in the paper should be run and compared against Centrifuger. As there are already a significant set of sequence analysis tools, any new method to introduce should be compared against a sufficient number of popular and established tools so the reader can see a broad overview of its performance.

In Results:

- Based on the results provided by authors themselves, it appears that on the selected datasets: Centrifuger is only beneficial for higher sensitivity at the species-level and for memory usage. Authors claimed that the underlying data structure of Centrifuger offers best performance for both time and space efficiency. In that case, why does Centrifuger runs 3 x more slowly than Centrifuge?
- Authors've run an analysis on real datasets (SARS-CoV-2 Oxford Nanopore WGS data) using Centrifuger. Were Centrifuge and Kraken2 also run on these data? How do the findings by Centrifuger compare to other tools? It's important to show these results so the reader can identify advantages/drawbacks of Centrifuger on real datasets compared to existing tools (especially regarding the presence of two clusters found? and execution performance?). Since the data here are sequencing data produced by a (very) different sequencing technology (Nanopore) than that of the simulated datasets (Illumina), the evaluation of the performance done earlier has no reason to be identical.

Open questions:

- I wonder what would be the performance (memory usage/classification speed/accuracy) of Centrifuger should the number of strain/assembly per species in the database increase?

Reviewer 4

Were you able to assess all statistics in the manuscript, including the appropriateness of statistical tests used? Yes, and I have assessed the statistics in my report.

Were you able to directly test the methods? Yes.

Comments to author:

The manuscript presents a comprehensive and clear exposition of the Run-Block compressed Burrows-Wheeler Transform (BWT), a promising data structure offering a beneficial space/query time tradeoff. It introduces 'Centrifuger', an advancement over 'Centrifuge' for taxonomic classification, demonstrating notable improvements in accuracy across various taxonomy levels, particularly at the species and genus levels, while maintaining a smaller index size. However, it is important to note that Centrifuger's assignment speed is somewhat slower compared to Kraken2, which is exceptionally fast.

Major Remarks:

Related Work: The manuscript introduces an innovative variant of full-text indexing but overlooks significant related works in bioinformatics, such as Spectral Burrows-Wheeler Transform, BOSS, Wheeler graph, and the move index. A comparative analysis or discussion of these related methods would provide a more comprehensive understanding of the proposed data structure's place within the broader field.

Comparative Analysis: While the improvement over Centrifuge for taxonomic classification is clear, the comparison is limited to its previous version and Kraken2. Including comparisons with other state-of-the-art tools like Ganon, Metamaps, and Taxor, presenting various time/memory tradeoffs, would offer a more robust evaluation of Centrifuger's performance.

Space/Time Tradeoff Clarity: The theoretical aspects of the hybrid run-length's space/time tradeoff are intriguing, yet the practical implications are somewhat ambiguous. Often, it appears to be larger in space and slower in query time compared to other data structures. A more detailed analysis or discussion on this aspect would be beneficial.

Figure 1 (Right Part) Clarity: The right part of Figure 1 currently does not effectively elucidate the data structure. Perhaps a more detailed or differently structured illustration could better aid in understanding.

Figure 2 Interpretation: Figure 2 is crucial in demonstrating the advantages of the new data structure. However, the differing behaviors of parts A and B make it challenging to extract a clear message. It is anticipated that the run-length BWT would be more compact at certain redundancy levels, but this is not evidently clear from the current representation. A revision or additional explanation might help clarify this expectation.

By addressing these points, the manuscript could provide a more thorough and contextualized understanding of the Run-Block compressed BWT and its implications in the field of bioinformatics.

Reviewer 5

Were you able to assess all statistics in the manuscript, including the appropriateness of statistical tests used? Yes, and I have assessed the statistics in my report.

Were you able to directly test the methods? Yes.

Comments to author:

In this manuscript, Song and Langmead present Centrifuger, an efficient taxonomic classification method that compares sequencing reads against a microbial genome database. The approach has multiple merits, and especially the run-block compressed data structure is definitely interesting, and its application on the BWT seems to bring multiple benefits.

Unfortunately, the benchmarking and testing approach seems not to follow best practice and to have the breadth and depth required to enable the Authors showcase the capabilities but also to communicate potential limitations to the community.

My detailed comments are presented in further detail below:

1. Mason is a surprising choice for a simulator due to its old age. Should be more than 10 years old. I'm not familiar whether it has been updated since then. There has been significant improvement during these 10-13 years in terms of sampling reads resembling actual sequencing data, error profiles matching more recent sequencers (probably it was only exposed to GAI1 at the time), etc. It's also impossible to understand the settings the Authors selected in terms of read length, and the other settings without installing the software. The relevant github page does not offer much information on the available settings and their effects.
2. The simulated dataset is quite small (1M sampled reads). I highly recommend testing using a dataset that resembles actual sizes occurring in the real world (between 20-100M reads), paired end, read lengths matching today's sequencers, and error rates. The whole point of a simulation is to simulate actual data and to provide to the Readers a potential expectation on how the algorithm will behave in a real-world scenario.
3. All time measurements are presented as ratios (e.g. number of reads /s). By using simulated datasets of sizes relevant to real-world situations (20-100M reads, PE), the Authors could also provide actual analysis time with 1 and more threads. These would help the Readers understand the feasibility of including this new approach in their analytical pipeline.
4. As the Authors mention, Centrifuger following the matching and scoring process reduces the number to within the threshold by promoting taxonomy IDs to their lowest common ancestor. What is the % of this event taking place in real world datasets? What is the % of reads that actually remain at the strain level when using a large index such as RefSeq in each simulated dataset?

5. It's not 100% clear to me how the 40-report threshold used in Centrifuger actually manifests in the results and tests. Are all 40 (or up to 40) reported as positive hits if the taxon is correct for all 40? I would expect only 1 to be counted as TP and probably that's what the Authors are performing but it's not clear from the text. Could the Authors add also false positives as well as a distance metric of their choice?

6. What is the ability of Centrifuger to derive quantifications from the metagenomic samples apart from taxonomic assignments? Meta-Kallisto, AGAMEMNON, Bracken, and other methods have moved beyond taxonomic assignments providing abundances down to species (Bracken) and strain (Meta-Kallisto, AGAMEMNON). How does Centrifuger compare against these methods at each level of taxonomic resolution (taxon quantifications, genus, species, and strain)? The TP and FP in these cases should not be at the common ancestor level but at the actual level of resolution (e.g. species or strain).

7. Could the Authors repeat the addition of *Legionella* and *Escherichia fergusonii* with a much larger number of randomly selected species/genera covering a large space of genome length and complexity?

8. The species-not-in test shows a potential weakness of the approach, which could affect its adoption in real world scenarios. In real world applications, the indexes are never complete, despite the community's best efforts. However, the impact might not be severe and should be quantified. Could the authors repeat the benchmarking/comparative tests by removing e.g. 25% and 50%, and 100% of the sampled genomes (a mix of missing strains, species, and genera)? Potentially larger number of reads might be needed to be sampled to capture the effect of missingness (50M reads or more). False positive rate might be of interest as well in these tests.

9. Can't the SARS-CoV-2 test be performed using strains and specific variant sequences as the index apart from the taxonomy ID? Couldn't Kraken2 be applied in this case? It would be of interest to show whether this segregation is captured or not when it is applied to this scenario.

10. The SARS-CoV-2 test should be quite interesting to repeat using the quantification methods as well (at strain level), since it should showcase Centrifuger's ability to go beyond what is currently possible with taxonomic rank approaches (and quantification methods).

Minor

1. Table S1 is currently clustered per approach. I would recommend the results to be clustered per metric and rank, so it's easy for the Reader to compare between the methods. The Authors could also mark with bold the best performing method per group of measurements.

Dear Editor,

We thank the reviewers for their objective assessments and constructive suggestions, which have helped us improve the study and clarify the manuscript. We also appreciate the opportunity to submit the revision, and would like to highlight the following major updates:

1. We incorporated two more classifiers, Ganon and KMCP, into the benchmarks on short reads.
2. We tested Kraken2, Centrifuge, Ganon, MetaMaps, and Taxor on the SARS-CoV-2 Oxford Nanopore WGS data and examined their sequence-level classifications.
3. We evaluated Centrifuger, Centrifuge, Kraken2, Ganon, and KMCP on a simulated data set generated by the simulator ART. We also benchmarked the five classifiers on a trimmed database where we excluded the true origins of the simulated reads.
4. We substantially revised the method section to clarify the classification procedure.
5. We expanded the discussion section of the manuscript to discuss about taxonomic profiling.

In the point-by-point response to all 5 reviewers, reviewer comments are in blue followed by our response in black. We hope that with all the reviewers' questions addressed, our revised manuscript is now suitable for publication at Genome Biology.

Sincerely,
Li Song and Ben Langmead

--

Reviewer #1: ===

The authors describe an extension of the already published tool Centrifuge. One main contribution of the work is the definition of a clever, and quite practical compression of the standard FM index used in Centrifuge, named the RunBlock compressed BWT. While the idea is quite standard, it was to my knowledge never published and makes this paper alone already worth publishing. The RBBWT simply divides the BWT into block of fixed length. Then the BWT is divided into two parts, the concatenation of blocks that have more than one character and the concatenation of blocks that consists of a single character, where per block only the one character is stored. Access to the BWT (or more important rank queries) then be emulated by three rank queries. One for the indicator bit vector and one each for the two strings.

This is a more practical solution than the RL encoded BWT, that is in theory optimal, but incurs quite a high run time penalty for rank queries and hence for exact search.. The authors then compare the run time

and space requirements of their implementation to the standard BWT, the RL compressed BWT and a hybrid solution.

As expected their run time is only about 3 times worse than the uncompressed version.

They also show that the resulting tool Centrifuger does perform similarly to Centrifuge and Kraken2. While it is not a full comparison (also to other tools like clark or ganon), it is sufficient to prove the claimed improvements of the RBBWT.

I have a number of minor comments that might improve the manuscript.

1) Please give more details of how the Sequence ID sampling is done. I assume that you mark text positions? If so, you would need to mark the beginning of an ID run in addition to the sampling rate markings. Please give more details how it is done.

We thank the reviewer for raising this concern. We have added more details about the sampling, where we explicitly mention the samplings correspond to the positions on the BWT sequence, i.e. the suffix array. With this implementation, we can check whether a given position in the BWT is a multiple of the sampling rate or not, with no need to explicitly mark the sampled positions. This strategy is also adopted in Bowtie and Centrifuge. The following is adapted from the revised "Taxonomic classification" section regarding the sampled sequence IDs:

Each match M corresponds to an interval on the BWT sequence, denoted as $[s_M, e_M]$, then we will resolve the sequence ID for each position on the BWT sequence between $[s_M, e_M]$, by applying LF-mapping. The LF-mapping procedure for resolving the position $p \in [s_M, e_M]$, terminates when it reaches a position p' on the BWT sequence with a sampled sequence ID, i.e., $p' \bmod 16 = 0$ with the default parameter. The sampled sequence ID at p' is the sequence ID corresponding to p .

2) its is usually a "sampled suffix array" not "subsampling"

3) it is "substrings" not "substrings"

We thank the reviewer for identifying these two typos. We have fixed them in the revised manuscript.

4) The performance in classification (sensitivity) differs between centrifuger and centrifuge. Why is that if you use the same search paradigm as in centrifuge? Can you elaborate more on the differences?

We agree that our manuscript should more clearly lay out the differences between Centrifuger and Centrifuge. While the search paradigm is the same between these two methods, there is a difference in how they resolve sequence IDs for a match found in many genomes. In Centrifuger, we will resolve a subset of the sequence IDs. Centrifuge, on the other hand, will simply skip all of them if the number of genomes containing this match is too large. The following is the more detailed description adapted from the revised method section:

A match M corresponds to an interval on the BWT sequence, with the interval denoted $[s_M, e_M]$. For a match that hits many sequences in the database, i.e., $m_e - m_s + 1$ is large, Centrifuger will resolve the sequence IDs for at most $40 \cdot \text{report_threshold}$ entries evenly distributed in the BWT interval. For example, if $m_e = 100$ and $m_s = 490$, then Centrifuger will resolve the 40 sequence IDs for positions at 100, 110, 120, ..., 480, and 490 on the BWT sequence with the default setting. Though this is a heuristic that can cause the algorithm to miss the true genome of origin, it is likely to generate scores for sequences in the same phylogeny branch and may help identify the correct taxonomy IDs at higher levels. This is the main difference between Centrifuge and Centrifuger in the classification stage, where Centrifuge ignores a match if $m_e - m_s + 1 > 200$ with the default setting.

Another example is from the newly added experiment about testing the accuracy of the classifiers on a trimmed database that has one genome per genus, which simulates the scenario where the read's true origin is not in the database. Since this database is relatively small, a match is likely to hit only a few genomes, and we observe Centrifuger and Centrifuge obtain almost identical performance (Figure S4A as

shown below). We also added this observation in the “Taxonomic classification” subsection of the Methods section.

Figure S4A. Performance of Centrifuger, Centrifuge, Kraken2, Ganon, and KMCP on the simulated data when classifying against a trimmed database that has one genome per genus and does not contain the true origins of the reads.

Fixing this bias in Centrifuge is non-trivial, as there are other places depend on this filter or threshold. Nevertheless, we do plan to address this issue in Centrifuge in the future.

===

Reviewer #2: ===

The paper further develops a metagenomics classification tool Centrifuge to become faster, more space-efficient, and accurate. This is done by replacing the FM-index of Centrifuge with an index that exploits run-length compressibility of the Burrows-Wheeler transform (BWT). Instead of taking an off-the-shelf tool like r-index for this purpose, the authors develop a new fixed-block scheme to partition BWT into run-blocks and non-run-blocks. They show that the optimal block size can be computed efficiently. Furthermore, the fixed-block scheme enables simpler computation of rank-queries to enable faster backwards search.

Authors first compare their approach experimentally to other FM-index implementations of which some exploit run-length compression. The space and time of the new index are competitive.

Then the authors move on to proposing some heuristic changes to Centrifuge and show that these changes improve its accuracy. Several experiments are conducted on variety of dataset to show improved accuracy against k-mer based indexes. However, the approach is still much slower than k-mer approaches limiting somewhat the scalability.

We thank the reviewer for this concern. We agree that Kraken2 is an exceptionally fast method. In the revised manuscript, we added Ganon and KMCP, two recent k-mer-based classifiers. On our simulated data set, we observed that Ganon and KMCP were slower than Centrifuger, while being substantially less sensitive than Centrifuger at lower taxonomy levels (Figure 1 shown below). This shows that though an FM-index-based classifier is slower than Kraken2, it is efficient for this task and brings huge improvements to the classification result.

Figure 3. Performance of Centrifuger, Centrifuge, Kraken2, Ganon, and KMCP on the simulated data generated from June 2023 RefSeq prokaryotic genomes (A) Sensitivity (left) and precision (right) of Centrifuger, Centrifuge and Kraken2 at various taxonomy ranks. (B) Peak memory usage of each classifier. (C) Classification speed of each classifier with a single thread.

The compression approach is very simple and easy to follow. However, authors refer to Figure 1 on many occasions, but that figure is very condensed and does not explain much. Important concepts like how the scores are defined for the matches are defined in confusing way. Basically, many concepts related to how Centrifuge works should be explained better.

We agree that we should do more to clarify Figure 1. We have added detailed explanations to the figure legend, which shall improve the readability for it (Figure 1 as shown below).

Figure 1. Overview of Centrifuger.

Left: classification procedure on the forward read. Centrifuger searches from the end of the read and applies the backward search to extend the match until reaching a mismatch. This yields the first 60-bp exact match hitting three sequences {X, Y, Z} in the database. Centrifuger then skips the mismatch and restarts the search again, giving the second 39-bp match hitting two sequences {X, Y}. The same search procedure applies to the reverse complement of the read. Centrifuger then scores each matched sequence and classifies the read to the sequences with the highest scores, where the example read is classified to the sequence X with the score 2601.

Right: the structure of Centrifuger’s lossless compressed FM-index. Centrifuger utilizes the RBBWT representation for the BWT sequence. In the example of compressing the BWT sequence “AAAAAGCTAAAA”, RBBWT represents it as two sequences “AA” and “ACGT” when the block size is 4. For the sequence IDs that are sampled on the BWT sequence, Centrifuger will compact their bits representation. In this example, there are four sequences in the database (W, X, Y, Z), so 2 bits are sufficient to represent the sequence ID. Therefore, for the substring of the BWT sequence shown in the example, Centrifuger spends 6 bits to represent sequence IDs that are sampled every other four positions on the BWT sequence.

Specific comments:

Background: You say "We previously developed... Centrifuge". However, [18] has many authors not in this submission, so I find "we" not suitable here. Maybe rephrasing "The first author co-developed...".

Page 9, line 4: "in bit is"  "in bits is"

We thank the reviewer for identifying these language issues, and we have changed them accordingly in the revised manuscript. Centrifuge’s code is based on Bowtie2, developed by the other author, Dr. Ben Langmead, of this manuscript, so we change the sentence to “We co-developed...”.

Page 14, line 13, Taxonomic classification: Please define what semi-maximal match means and explain how you find them using backward search. You refer to Figure 1, but it does not help in figuring out what is happening here.

We explicitly define the semi-maximal match in the revised manuscript in the “Method overview” section and the “Taxonomic classification” section. For example, in the “Method overview” section, we added: “We call the match semi-maximal because only one end of the match cannot be extended further.”

... line 19: "Each valid match"  "Each valid match of length ℓ ".

We thank the reviewer for this suggestion. We have revised this part as "Each valid match M of length l_M will..."

.. lines 24-26: The definitions do not seem to make any sense. What do you mean by " i in m means match m hits.."? There are many undefined terms here, like ℓ_m and notation i in m .

We appreciate the reviewer for identifying these definition issues. We have revised the "Taxonomic classification" section to clarify them. We have added the subscript M for the properties that relate to a match M , so the length of a match is described as "Let l_M denote the length of a match M ". The revised sentence for the scoring function is now "For each resolved sequence ID, Centrifuger will sum up its scores across the matches using the formula $\text{score}(\text{sequence } I) = \sum_{M \in \text{sequence } I} (l_M - 15)^2$, where $M \in \text{sequence } I$ means the match M is found in the sequence with ID I ."

===

Reviewer #3: Song and Langmead presents Centrifuger, a new microbial sequence classification program, defined as an improvement of Centrifuge, a previous method published in 2016.

The manuscript is well written, however I'd like to bring some issues to the authors' attention:

Minor issues:

- In Background, end of 2nd paragraph: definition of n for $O(n)$ is missing.
- In Background, last sentence of the 3rd paragraph: "Inspired by the observation" -> "Inspired by this observation".

We appreciate the reviewer for finding these issues. We have updated the manuscript according to the suggestions.

- In Results, authors wrote "we downloaded the first 10 simulated samples from the CAMI2 Challenge datasets at https://frl.publisso.de/data/frl:6425521/strain/short_read/". But what is specifically the order used to select these first 10 (Based on file size? date modified? lexicographic? etc.)? Please list the IDs/sample names so we can reproduce your results.

We mention that they are the first 10 samples in lexicographic order now. The revised manuscript for this part is: "In addition to our own simulated data, we downloaded the 10 simulated samples from the CAMI2 Challenge datasets at https://frl.publisso.de/data/frl:6425521/strain/short_read/ by lexicographical order, i.e. sample_0 to sample_9."

Major issues:

In Methods:

- Why do the authors compare Centrifuger against only 2 other methods? I understand these 2 methods come from the same lab/university thus I'd recommend to include methods from other research groups to present a comprehensive comparison. Methods like MetaPhlan2, CLARK, Ganon and KMCP that are cited in the paper should be run and compared against Centrifuger. As there are already a significant set of sequence analysis tools, any new method to introduce should be compared against a sufficient number of popular and established tools so the reader can see a broad overview of its performance.

We appreciate the reviewer’s concern about the methods included in the comparisons. In the revised manuscript, we incorporated Ganon and KMCP in the evaluations. MetaPhlan only stores marker gene sequences, which will miss all the simulated reads from non-marker gene regions. CLARK is very memory-demanding and cannot efficiently store more recent microbial genome data. Therefore, we excluded MetaPhlan and CLARK in the evaluations. In our evaluations, Centrifuger showed much better performance at the species and genus levels than Ganon and KMCP (Figure 3A as shown above in the response to Reviewer 2). Furthermore, Centrifuger was much more computational efficient than the two recent classifiers (Figure 3B,C as shown above in the response to Reviewer 2).

In Results:

- Based on the results provided by authors themselves, it appears that on the selected datasets: Centrifuger is only beneficial for higher sensitivity at the species-level and for memory usage. Authors claimed that the underlying data structure of Centrifuger offers best performance for both time and space efficiency. In that case, why does Centrifuger runs 3 x more slowly than Centrifuge?

We thank the reviewer for raising these questions. We focused on the sensitivity at the species level and the genus level because the performance of the tested classifiers was the most divergent at these two ranks. In our evaluations, Centrifuger achieved the best classification accuracy across all the taxonomy ranks (Figure 3A as shown above in the reply to Reviewer 2). For another example, in the revised manuscript, we also calculated the F1 score ($2 * \text{sensitivity} * \text{precision} / (\text{sensitivity} + \text{precision})$) for the evaluations on the CAMI2’s simulated data. Centrifuger achieved the highest scores across the six taxonomy ranks (Figure S5 as shown below).

Figure S5. F1 scores of Centrifuger, Centrifuge, Kraken2, Ganon, and KMCP at various taxonomy ranks on the 10 simulated data sets from CAMI2

For the question about Centrifuger being 3x slower than Centrifuge, this is due to the RBBWT structure. Though it is more efficient than other lossless compression representations, it is still slower than the uncompressed representation used in Centrifuge. We added this sentence to the “Performance on classifying simulated data” section to clarify: “Centrifuger was about three times slower than Centrifuge, reflecting the earlier observation that the rank query on RBBWT was three times slower than on an uncompressed data structure.”

- Authors've run an analysis on real datasets (SARS-CoV-2 Oxford Nanopore WGS data) using Centrifuger. Were Centrifuge and Kraken2 also run on these data? How do the findings by Centrifuger compare to other tools? It's important to show these results so the reader can identify advantages/drawbacks of

Centrifuger on real datasets compared to existing tools (especially regarding the presence of two clusters found? and execution performance?). Since the data here are sequencing data produced by a (very) different sequencing technology (Nanopore) than that of the simulated datasets (Illumina), the evaluation of the performance done earlier has no reason to be identical.

We appreciate the reviewer for raising this concern about the evaluations on the Oxford Nanopore data. In the revised manuscript, we included Centrifuge, Kraken2, Ganon (with a different parameter setting), MetaMaps, and Taxor for the application on long reads. MetaMaps was very memory demanding and could not load the database that contained RefSeq prokaryotic and human genomes as in Centrifuge. Taxor generated a different taxonomy tree structure when creating the index, such as skipping the SARS-CoV-2's taxonomy ID and directly putting them to the ID 694009. Therefore, we created the indices for MetaMaps and Taxor just on the 93 SARS-CoV-2 genomes. Even though MetaMaps and Taxor classified reads against the SARS-CoV-2-only database, their classification rates were lower than other methods. The evaluation results in the revised manuscript are as follows:

“Centrifuge classified about 99.95% of all the input reads to the taxonomy ID 2697049 on average, while Kraken2 and Ganon were slightly less sensitive and classified 99.90% and 98.96% reads to the ID 2697049 on average, respectively. Centrifuge had a slightly different LCA search implementation, so it classified 98.90% of the reads to either the ID 2697049 or ID 694009, where ID 694009 was the parent node of 2697049. Despite using a SARS-CoV-2-only database, MetaMaps and Taxor only classified 78.04% and 70.91% reads on average, respectively. The lower sensitivity for MetaMaps was mainly due to its default setting of skipping reads shorter than 1000 bp, while it mapped almost all the remaining reads of sufficient length. When looking at the sequence-level classification, Centrifuge assigned 23.7% of the input reads with unique sequence IDs across all the samples. For the other tested methods, only Centrifuge, Ganon and Taxor reported sequence-level classifications, but they uniquely classified 15.1%, less than 0.1% and about 0.1% of the reads, respectively. “

Besides Centrifuge, only Centrifuge reported enough sequence-level classifications, and it also found the variant fraction in the samples from the two projects were separated (Figure S7B as shown below).

Figure S7B. The cluster heatmap for read fractions from Centrifuge in the SARS-CoV-2 sequence-level analysis. The rows are SARS-CoV-2 variants present in the RefSeq and GenBank, and the columns are the Oxford Nanopore WGS samples. The row values are z-score transformed.

Perhaps there is no need for many WGS reads in virus analysis, the number of reads in the samples from the two projects was small, typically less than 30K in PRJEB40277 and less than 200K in PRJNA673096. As

a result, the tested methods could finish a sample within 10 minutes using eight threads and the database loading time took a significant proportion of the running time. Therefore, we did not include the running time in the revised manuscript.

Open questions:

- I wonder what would be the performance (memory usage/classification speed/accuracy) of Centrifuger should the number of strain/assembly per species in the database increase?

For memory usage, Centrifuger's index size still grows linearly as the database size increases, which we acknowledged in the original Discussion section. Specifically, though RBBWT is a sublinear-space data structure, the sampled suffix information/taxonomy IDs take linear space if the sampling rate remains constant as the database increases. The overall space for the sampled taxonomy IDs is fewer than an uncompressed BWT or the RBBWT, so Centrifuger's index size will increase linearly with a smaller coefficient compared to a conventional FM-index. Furthermore, the user can adjust the sampling rate to reduce the index size at the expense of running time, whereas the BWT is much harder to reduce, which motivates us to design the RBBWT representation.

The classification speed is not strongly impacted as the database size increases. The classification procedure in Centrifuger can be split into two stages. The first step is to find matches, which is backward searches on the FM-index. The time complexity of this step depends on the read length. The second step is to resolve the sequence IDs for a match. When an exact match can be found in many sequences in the database, Centrifuger will only resolve a subset of them (40 with the default value). Combining the two steps, we expect Centrifuger's speed will slow down when the number of strains per species in the database starts to increase at the beginning. But as the growth continues, the speed should level off. This is the hypothesis given the sampling rate of the taxonomy IDs in the FM-index remains the same. In our discussion section, we provide an example of the tradeoff between running time and space usage as follows:

“Nevertheless, the overall space complexity of Centrifuger is still $O(n)$ words due to the structure of sampled sequence IDs, making it worse than r -index's $O(r)$ words in the future as the repetitiveness of the genome database may grow fast. Though users can select a sparser sampling rate to maintain the space usage, this is at the expense of time efficiency. For instance, the index size became 32 GB when increasing the sampling rate from the default 16 to 32, but the classification speed of a thread decreased from 163K read/minute to 102K read/minute. Future work is needed to design a representation of the sampled sequence ID in sublinear space without sacrificing classification speed for microbial genomes. “

For classification accuracy, we expect that Centrifuger may lose the ability for strain-level classifications as the number of strains per species increases to a certain degree. But will expect it will maintain high accuracy for the species-level and genus-level classification.

Reviewer #4: The manuscript presents a comprehensive and clear exposition of the Run-Block compressed Burrows-Wheeler Transform (BWT), a promising data structure offering a beneficial space/query time tradeoff. It introduces 'Centrifuger', an advancement over 'Centrifuge' for taxonomic classification, demonstrating notable improvements in accuracy across various taxonomy levels, particularly at the species and genus levels, while maintaining a smaller index size. However, it is important to note that Centrifuger's assignment speed is somewhat slower compared to Kraken2, which is exceptionally fast.

Major Remarks:

Related Work: The manuscript introduces an innovative variant of full-text indexing but overlooks significant related works in bioinformatics, such as Spectral Burrows-Wheeler Transform, BOSS, Wheeler graph, and the move index. A comparative analysis or discussion of these related methods would provide a more comprehensive understanding of the proposed data structure's place within the broader field.

We thank the reviewer for these suggestions. The move structure is relevant to r-index, so we added it to the Introduction section. For the spectral Burrows-Wheeler transformation, which contains a variation of the BOSS structure, we checked the software Themisto (Alanko et al., 2023), a method for pseudoalignment based on the spectral BWT. When indexing the RefSeq prokaryotic genomes, Themisto yielded larger index than Centrifuger, where the k-mer representation without colors (just .tdbg file) was already larger than Centrifuger's index. We did not count the color file size because the color structure may be different if sBWT would be extended for the taxonomic classification. We added this experiment to the revised manuscript in the "Performance on classifying simulated data" section as following:

"Methods like Kraken2, Ganon and KMCP reduces the memory usage by discarding k-mer information. We also explored the space usage of succinct colored k-mer representations [32], which can keep all the k-mer information along with their color (sequence ID) information. We created the index on these RefSeq prokaryotic genomes using Themisto v3.2.1 [33], a pseudoalignment method based on the spectral BWT [34], using a k-mer size of 31. Its index, without the color component, took 44 GB space (the .tdbg file), which was already more than Centrifuger's 41-GB full index size. Since Themisto is not designed for taxonomic classification, we excluded it from other evaluations. Nevertheless, this observation suggests that succinct colored k-mer representations could be memory-efficient enough for read classifications against a large microbial genome database."

The Wheeler graph-based representation requires variants calling on the genomes first and may need extensive adjustment or preprocessing to work with the microbial genome database. Therefore, we did not include it in this manuscript.

Comparative Analysis: While the improvement over Centrifuge for taxonomic classification is clear, the comparison is limited to its previous version and Kraken2. Including comparisons with other state-of-the-art tools like Ganon, MetaMaps, and Taxor, presenting various time/memory tradeoffs, would offer a more robust evaluation of Centrifuger's performance.

We thank the reviewer for this suggestion. We have included Ganon and KMCP in the short-read evaluations in the revised manuscript. Centrifuger showed significantly better classification results than these two methods and was also more time- and memory-efficient (Figure 3 as shown above in the response to Reviewer 2). MetaMaps and Taxor are for long-read platforms. We evaluated them in the exploration of the SARS-CoV-2 Oxford Nanopore WGS data, but their classification rates were lower than Centrifuge's (see the above response to the second last question of Reviewer 3).

Space/Time Tradeoff Clarity: The theoretical aspects of the hybrid run-length's space/time tradeoff are intriguing, yet the practical implications are somewhat ambiguous. Often, it appears to be larger in space and slower in query time compared to other data structures. A more detailed analysis or discussion on this aspect would be beneficial.

We appreciate the reviewer for this suggestion. Our original manuscript has discussed the space tradeoff as: “Hybrid run-length compressed BWT uses the same amount of space as the wavelet tree representation when the repetitiveness of the sequence is low, and its space usage converges to the RLBWT’s when the repetitiveness grows.” in the “Hybrid run-length compression” section. We now explicitly mention that this representation is slower than RLBWT for rank queries: “The rank query on the hybrid run-length compressed sequence is like the run-block compression, where we combine the ranks from T_R and T_P , making it slower than the rank query on RLBWT”. We also added a sentence to describe its practical value as: “Despite being less computationally efficient than other representations, the hybrid run-length compressed BWT is flexible and allows methods to handle various texts with a wide range of repetitiveness without altering the underlying data structure.”.

We designed and included hybrid run-length compression in the manuscript because it was a natural extension of RLBWT when the average run length was low. Then we showed that RBBWT was a better data structure for this scenario.

Figure 1 (Right Part) Clarity: The right part of Figure 1 currently does not effectively elucidate the data structure. Perhaps a more detailed or differently structured illustration could better aid in understanding.

We thank the reviewer for raising this concern. We have added a detailed description to the figure legend (Figure 1 shown above in the response to Reviewer 2) to elucidate the data structure.

Figure 2 Interpretation: Figure 2 is crucial in demonstrating the advantages of the new data structure. However, the differing behaviors of parts A and B make it challenging to extract a clear message. It is anticipated that the run-length BWT would be more compact at certain redundancy levels, but this is not evidently clear from the current representation. A revision or additional explanation might help clarify this expectation.

We appreciate the reviewer for this suggestion. We agree that the two different trends will be confusing. In the revised manuscript, we moved part A to the supplementary figures and only kept part B in the main figure (Figure 2 and Figure S1 as shown below).

Figure 2. Computational efficiency of the wavelet tree, RLBWT, hybrid run-length compression, and RBBWT (A) Bits used to represent one base pair (bp) as the average run length of the BWT sequence (n/r) increases when representing increasingly more genomes with species ID 564 (*Escherichia fergusonii*). (B) Bits used to represent one bp when representing genomes with genus ID 445 (*Legionella*). (C) Rank query time.

Figure S1. Space usage of the wavelet tree, RLBWT, hybrid run-length compression and RBBWT when adding genomes with (A) species *Escherichia fergusonii* (taxonomy ID 564) and (B) the genus *Legionella* (taxonomy ID 445)

Furthermore, we conducted the space-changing experiment on the species *Chlamydia trachomatis* that was known to have highly redundant genomes. The analysis showed that run-length BWT and RBBWT's space usage started to separate when the average run length was about 20 (Figure S2A shown below). In the revised manuscript, we also wrote "We also compared the space usage of the BWT representations by adding the genomes from the species *Chlamydia trachomatis* (taxonomy ID 810) whose strains had highly similar sequences [18]. Again, RBBWT was the most memory-efficient data structure when l was less than or around 10 (Figure S2A as shown below). For this species, l reached 56.0 after adding all the genomes, and RBBWT's space was about a quarter of the uncompressed wavelet tree's and twice as much as RLBWT's in this case."

Figure S2A. Space usage of the wavelet tree, RLBWT, hybrid run-length compression and RBBWT when adding genomes with species *Chlamydia trachomatis* (taxonomy ID 813) Left: absolute space usage in megabytes (MB). Right: bits used to represent one base pair (bp) when the average run length of the BWT sequence (n/r) increases.

By addressing these points, the manuscript could provide a more thorough and contextualized understanding of the Run-Block compressed BWT and its implications in the field of bioinformatics.

===

Reviewer #5:

In this manuscript, Song and Langmead present Centrifuger, an efficient taxonomic classification method that compares sequencing reads against a microbial genome database. The approach has multiple

merits, and especially the run-block compressed data structure is definitely interesting, and its application on the BWT seems to bring multiple benefits.

Unfortunately, the benchmarking and testing approach seems not to follow best practice and to have the breadth and depth required to enable the Authors showcase the capabilities but also to communicate potential limitations to the community.

My detailed comments are presented in further detail below:

1. Mason is a surprising choice for a simulator due to its old age. Should be more than 10 years old. I'm not familiar whether it has been updated since then. There has been significant improvement during these 10-13 years in terms of sampling reads resembling actual sequencing data, error profiles matching more recent sequencers (probably it was only exposed to GAIL at the time), etc. It's also impossible to understand the settings the Authors selected in terms of read length, and the other settings without installing the software. The relevant github page does not offer much information on the available settings and their effects.

We appreciate the reviewer's concern about using Mason. In the revised manuscript, we generated another simulated data set with one million read pairs using ART v2.5.8, which was updated on 2016-06-05. We used the Illumina HiSeq 2500's error profile for ART. On this ART-generated data, Centrifuger remained the top performance (Figure S3 as shown below).

Figure S3. Sensitivity (left) and precision (right) of Centrifuger, Centrifuge, Kraken2, Ganon, and KMCP on the simulated data generated from June 2023 RefSeq prokaryotic genomes using ART

We also explicitly mention the key properties of the two simulated data set in the “Performance on classifying simulated data” section of the revised manuscript as following:

For Mason:

“We compared Centrifuger, Centrifuge, Kraken2, Ganon, and KMCP’s accuracy on one million 100-base-pair (bp) paired-end short reads simulated by Mason [30] from 34,190 prokaryotic genomes (RefSeq bacteria+archaea). We set the sequencing error rate in Mason to be 1%, a value that was higher than the Illumina sequencing platform, to mimic the microbial genome variations in real data analysis.”

For ART:

“In addition to Mason, we compared the five classifiers on another set of one million 100-bp paired-end short reads simulated by ART [31].”, and “..., where ART’s default error rate was around 0.15%.”

2. The simulated dataset is quite small (1M sampled reads). I highly recommend testing using a dataset that resembles actual sizes occurring in the real world (between 20-100M reads), paired end, read lengths matching today’s sequencers, and error rates. The whole point of a simulation is to simulate actual data and to provide to the Readers a potential expectation on how the algorithm will behave in a real-world scenario.

We agree with the reviewer on the realness of the simulated data. This motivated us to utilize the simulated data sets from the Critical Assessment of Metagenome Interpretation 2 (CAMI2) benchmark study (Meyer et al., 2022, Nature Methods, cited more than 150 times so far) in the original manuscript. This simulated data set was generated around 2019. In the revised manuscript, we added more details about this data set in the “Performance on classifying simulated data” section, such as “Each sample has about 6.7 million 150-bp read pairs.” In this evaluation, Centrifuger still showed the highest sensitivity at the species and genus levels with comparable precision as other methods (revised Figure 4 as shown below; F1 score as Figure S5 is shown above in the response to Reviewer 3).

Figure 4. Sensitivity (top) and precision (bottom) of Centrifuger, Centrifuge, Kraken2, Ganon, and KMCP at various taxonomy ranks on the 10 simulated data sets from CAMI2

We kept the Mason and ART generated data to be 100 bp pair-end, so the classifiers could be tested on diverse scenarios. Though the number of reads in a sample is still lower than what the reviewer suggested, we believe one million or 6.7 million reads in a sample is sufficient to demonstrate the high sensitivity and good precision of Centrifuger. In the revised manuscript, we evaluated about 69 million simulated read pairs in total.

3. All time measurements are presented as ratios (e.g. number of reads /s). By using simulated datasets of sizes relevant to real-world situations (20-100M reads, PE), the Authors could also provide actual analysis time with 1 and more threads. These would help the Readers understand the feasibility of including this new approach in their analytical pipeline.

We thank the reviewer for this suggestion. We used the 1M reads for time measurement with a single thread so that the experiments could be finished and replicated multiple times within a reasonable time. The single-thread running time can also reflect the underlying data structure's efficiency and is less affected by I/O bandwidth and parallelization engineering. We agree that this number of reads is too low to reflect real data. Therefore, in the revised manuscript, we concatenated the 10 CAMI simulated data sets to create a data set with 67 million 150-bp paired-end reads. We compared Centrifuger, Centrifuge, Kraken2, Ganon, and KMCP's running times on this data set with 8 threads. Our observation in the revised manuscript is the following:

"We concatenated the 10 samples into a large data set containing about 67 million read pairs to compare the speed of the classifiers running with eight threads. Kraken2 was the fastest method (finished in about 7min), followed by Centrifuge (42min) and Centrifuger (1h35min). With multithreading, Centrifuger was about 2.26 times slower than Centrifuge, reducing the threefold speed difference when running on a single thread. Ganon was more time efficient on the CAMI2 data set (2h40min) than on the Mason-generated simulated data, which could be due to its low sensitivity and might skip many read classifications. KMCP did not scale well and took more than 18 hours to finish."

4. As the Authors mention, Centrifuger following the matching and scoring process reduces the number to within the threshold by promoting taxonomy IDs to their lowest common ancestor. What is the % of this event taking place in real world datasets? What is the % of reads that actually remain at the strain level when using a large index such as RefSeq in each simulated dataset?

We appreciate the reviewer for raising this question. We showed this percentage in the original Figure 3A (revised version shown above in the response to Reviewer 2), where we observed less than 25% of reads remain at the strain level. A similar trend is observed on the simulated data generated by ART (Figure S3 as shown above). In the revised manuscript, we added the sentence: "In both simulated data sets, the sensitivity at the strain level was very low (<25%) for all five methods, suggesting that most reads cannot be uniquely assigned to a strain in the RefSeq prokaryotic genome database."

5. It's not 100% clear to me how the 40-report threshold used in Centrifuger actually manifests in the results and tests. Are all 40 (or up to 40) reported as positive hits if the taxon is correct for all 40? I would expect only 1 to be counted as TP and probably that's what the Authors are performing but it's not clear from the text. Could the Authors add also false positives as well as a distance metric of their choice?

The 40*report_thread is to restrain the number of resolved sequence IDs for each match, as some of the matches can hit hundreds of sequences and will take a long time to resolve all of them. The following is adapted from the "Taxonomic classification" section as a concrete example:

A match M corresponds to an interval on the BWT sequence, where the interval can denote as $[s_M, e_M]$. For a match that hit many sequences in the database, i.e., $m_e - m_s + 1$ is large, Centrifuger will resolve the sequence IDs for at most 40-report_threshold entries evenly distributed in the BWT interval. For example, if $m_e = 100$ and $m_s = 490$, then Centrifuger will resolve the 40 sequence IDs for positions at 100, 110, 120, .., 480, and 490 on the BWT sequence with the default setting.

In the final classification result output, by default, Centrifuger outputs the LCA of the highest-score taxonomy IDs as in Kraken2. We revised the last part of the "Method overview" section for clarification as the following:

“When the number of reported IDs exceeds the user-specified threshold (default report threshold 1), Centrifuger reduces the number to within the threshold by promoting some taxonomy IDs to their lowest common ancestors (LCAs) in the taxonomy tree. In other words, Centrifuger reports the LCA of the taxonomy IDs for a read by default, as in Kraken2.”

6. What is the ability of Centrifuger to derive quantifications from the metagenomic samples apart from taxonomic assignments? Meta-Kallisto, AGAMEMNON, Bracken, and other methods have moved beyond taxonomic assignments providing abundances down to species (Bracken) and strain (Meta-Kallisto, AGAMEMNON). How does Centrifuger compare against these methods at each level of taxonomic resolution (taxon quantifications, genus, species, and strain)? The TP and FP in these cases should not be at the common ancestor level but at the actual level of resolution (e.g. species or strain).

We appreciate the reviewer for raising the question about quantification. Centrifuger is for taxonomic classification and does not have a quantification module. While the EM algorithm is commonly used in quantification to handle multi-classified reads, such as in Centrifuge, there are newly proposed methods to improve the profiling accuracy by adjusting the EM procedure. For example, AGAMEMNON (Skoufos et al, 2022), a method suggested by the reviewer, adds a step between the EM iterations to handle genomes that have no unique assigned reads. Another new method on Biorxiv, Centrifuge+ (Liu et al., 2023), modifies the likelihood function in the EM algorithm to consider the unique mapping rate of a species. Many methods, like Bracken, Ganon, KMCP, and Taxor, adopt a two-step strategy, where they conduct taxonomic classification first and then use the output for quantification. It is possible to make Centrifuge’s output compatible with the other tools’ quantification modules. Nevertheless, we still need to conduct comprehensive evaluations to find the appropriate quantification method that we can either integrate into Centrifuger or recommend as the following step following Centrifuger. Such evaluations are beyond the scope of this study. We have added a paragraph in the Discussion section to discuss about the quantification as following:

“Besides taxonomic classification, another important problem in metagenomic data analysis is taxonomic profiling, i.e., determining the abundance for each species or at user-specified taxonomic ranks. It is feasible to profile the abundances directly without taxonomic classification for reads, such as in Meta-Kallisto [43] and Sylph [44]. However, many profiling methods still require taxonomic classification results which can identify species of low abundance with few reads supporting them. For instance, Bracken uses Kraken2’s input for taxonomic profiling with a Bayesian method [45]. Ganon, KMCP, and Taxor, which are benchmarked in this work, need to conduct taxonomic classifications before profiling. Centrifuge integrates the abundance estimation based on the Expectation-Maximization (EM) algorithm [46] internally. After the release of Centrifuge, some methods, like AGAMEMNON [47] and Centrifuge+ [48], improve the profiling accuracy by adjusting the likelihood function and the EM algorithm procedure. We still need to systematically compare these profiling techniques and either integrate them into Centrifuger or make Centrifuger’s output compatible with these methods in the future. For example, Centrifuger provides the script to summarize the classification results into a Kraken-style report file that can serve as the input for Bracken.”

7. Could the Authors repeat the addition of *Legionella* and *Escherichia fergusonii* with a much larger number of randomly selected species/genera covering a large space of genome length and complexity?

We thank the reviewer for the suggestion of exploring RBBWT for other species. *Legionella* and *Escherichia fergusonii* are among the top 50 genera and species with the most number of genomes, respectively. We believe they also represent a typical genome complexity. Therefore, in the revised

manuscript, we added the experiment on the species *Chlamydia trachomatis* and its genus *Chlamydia*, where this species' genome sequences are highly similar to each other as found in our previous study in Centrifuge. *Chlamydia* has more genomes than *Legionella*. Though *Chlamydia trachomatis* has fewer number of genomes than *Escherichia fergusonii*, it is among the top 100 species with the greatest number of genomes. In these experiments, RBBWT was still the most memory-efficient method when the average run length was around 10 (Figure S2A for the species *Chlamydia trachomatis* as shown above in the response to Reviewer 4; Figure S2B for genus *Chlamydia* as shown below). When the average run length was large, such as 56.0, RBBWT's space was about a quarter of the uncompressed wavelet tree's and twice as much as RLBWT's in this case. This analysis also helped us examine the space usage of RBBWT on highly repetitive genomes.

Figure S2B. Space usage of the wavelet tree, RLBWT, hybrid run-length compression and RBBWT when adding genomes with species *Chlamydia* (taxonomy ID 810) Left: absolute space usage in megabytes (MB). Right: bits used to represent one base pair (bp) when the average run length of the BWT sequence (n/r) increases.

8. The species-not-in test shows a potential weakness of the approach, which could affect its adoption in real world scenarios. In real world applications, the indexes are never complete, despite the community's best efforts. However, the impact might not be severe and should be quantified. Could the authors repeat the benchmarking/comparative tests by removing e.g. 25% and 50%, and 100% of the sampled genomes (a mix of missing strains, species, and genera)? Potentially larger number of reads might be needed to be sampled to capture the effect of missingness (50M reads or more). False positive rate might be of interest as well in these tests.

The reviewer raises a very insightful question about the effects of missing true genomes. In practice, the fraction of missing genomes, e.g. 25%, 50%, is unknown. Therefore, we created a simulated dataset where 100% of the true genomes were missing, and the other fractions can be interpolated from the 0%-missing's and 100%-missing's results. In brief, we randomly selected one genome per genus from the RefSeq prokaryotic genomes, and then excluded the reads from these selected genomes in the Mason-generated simulated data. Centrifuger, Centrifuge, and Kraken2 had very similar performance in the missing-genome evaluation, where they were much more sensitive than Ganon and KMCP, but were less precise (Figure S4A as shown above in the response to Reviewer 2). When looking at the F1 score, Centrifuger, Centrifuge, and Kraken2 achieved the best results (Figure S4B as shown below).

Figure S4B. F1 score ($(2 * \text{sensitivity} * \text{precision} / (\text{sensitivity} + \text{precision}))$) of Centrifuger, Centrifuge, Kraken2, Ganon, and KMCP on the simulated data when classifying against a trimmed database that has one genome per genus and does not contain the true origins of the reads

This is also in line with the revised species-not-in test, where we incorporated Ganon and KMCP in the analysis (Figure 5B as shown below), and Kraken2 and Centrifuger achieved higher F1 scores than other classifiers (Figure S6 as shown below).

Figure 5B. Performance of Centrifuger, Centrifuge, Kraken2, and Ganon on bacterial WGS data sets Sensitivity (left) and precision (right) if species of the bacteria are not in the database but their genera are present in the database.

Figure S6. F1 scores of Centrifuger, Centrifuge, Kraken2, Ganon and KMCP on bacterial WGS data sets, where the species of the bacteria are not in the database but their genera are present in the database

9. Can't the SARS-CoV-2 test be performed using strains and specific variant sequences as the index apart from the taxonomy ID? Couldn't Kraken2 be applied in this case? It would be of interest to show whether this segregation is captured or not when it is applied to this scenario.

Kraken2 only reports the taxonomy ID in the result even if we do put the strains and specific variant sequences separately when creating the index. One could change the internal procedure of Kraken2 and create a dummy taxonomy ID for each input strain. However, this could not be a common practice and may break other parts of Kraken2. In the revised manuscript, we included Centrifuge, Ganon, and Taxor for this analysis. Centrifuge uniquely classified 15.1% of the input reads to a SARS-CoV-2 variant, while Ganon and Taxor only uniquely classified about 0.01% and 0.1% of the reads. We observed a similar segregation between the two projects' samples using Centrifuge's result (Figure S7B as shown above in the response to Reviewer 2). Due to the low sequence-level classification rate, we could not reproduce the segregation analysis using Ganon and Taxor's output. Because Centrifuge and Centrifuger had similar classification procedures, the analysis could still be biased. Therefore, we added that the segregation pattern could be from batch effects.

10. The SARS-CoV-2 test should be quite interesting to repeat using the quantification methods as well (at strain level), since it should showcase Centrifuger's ability to go beyond what is currently possible with taxonomic rank approaches (and quantification methods).

We have applied the methods Ganon and Taxor to the SARS-CoV-2 data set, where the two methods also report sequence-level classifications and have modules for quantification. However, these two methods generated too few sequence-level classification results, with only around 0.1% of the reads uniquely assigned to a sequence. This suggests that their quantifications will not provide insight to the sequence-level analysis. The quantification result from Centrifuge is limited to taxonomy IDs, so it cannot provide information to the sequence-level abundances either.

Minor

1. Table S1 is currently clustered per approach. I would recommend the results to be clustered per metric and rank, so it's easy for the Reader to compare between the methods. The Authors could also mark with bold the best performing method per group of measurements.

We appreciate the reviewer for this suggestion. We have revised Table S1 (shown below) following the suggestions.

rank	Matched (TP)	Predicted (P)	Cases (T)	sensitivity	precision	method
strain	231270	241184	1000000	0.2313	0.9589	Centrifuger
	229849	441444	1000000	0.2298	0.5207	Centrifuge
	221359	232119	1000000	0.2214	0.9536	Kraken2
	185290	186100	1000000	0.1853	0.9956	Ganon
	173246	173579	1000000	0.1732	0.9981	KMCP
species	816975	832477	1000000	0.817	0.9814	Centrifuger
	607293	618127	1000000	0.6073	0.9825	Centrifuge
	607231	626061	1000000	0.6072	0.9699	Kraken2

	504921	506452	1000000	0.5049	0.997	Ganon
	378273	378309	1000000	0.3783	0.9999	KMCP
genus	967470	972537	999521	0.9679	0.9948	Centrifuger
	765665	771680	999521	0.766	0.9922	Centrifuge
	913546	918606	999521	0.914	0.9945	Kraken2
	762842	763340	999521	0.7632	0.9993	Ganon
	853067	853070	999521	0.8535	1.0000	KMCP
family	992539	992922	997389	0.9951	0.9996	Centrifuger
	782725	785742	997389	0.7848	0.9962	Centrifuge
	988327	988860	997389	0.9909	0.9995	Kraken2
	830304	830345	997389	0.8325	1.0000	Ganon
	969460	969460	997389	0.972	1.0000	KMCP
order	995976	996205	998781	0.9972	0.9998	Centrifuger
	786747	788930	998781	0.7877	0.9972	Centrifuge
	993684	994021	998781	0.9949	0.9997	Kraken2
	834722	834745	998781	0.8357	1.0000	Ganon
	980796	980796	998781	0.982	1.0000	KMCP
class	996386	996466	997481	0.9989	0.9999	Centrifuger
	787116	788637	997481	0.7891	0.9981	Centrifuge
	994895	995050	997481	0.9974	0.9998	Kraken2
	835781	835785	997481	0.8379	1.0000	Ganon
	985743	985743	997481	0.9882	1.0000	KMCP
phylum	999402	999448	999749	0.9997	1.0000	Centrifuger
	790843	791705	999749	0.791	0.9989	Centrifuge
	998390	998475	999749	0.9986	0.9999	Kraken2
	838708	838708	999749	0.8389	1.0000	Ganon
	990439	990439	999749	0.9907	1.0000	KMCP

Table S1. The classification accuracy at various taxonomy ranks in the Mason-generated simulated data. Sensitivity=TP/T, Precision=TP/P. The highest values of sensitivity and precision at each rank are bolded.

**Figure 3A.** Sensitivity (left) and precision (right) of Centrifuger, Centrifuge, Kraken2, Ganon, and KMCP on the simulated data generated from June 2023 RefSeq prokaryotic genomes

2) ganon achieves best results with the EM-algorithm (which is the default) but was changed to LCA ("--multiple-matches lca"). Why?

We thank the reviewer for raising this question. We agree that the reassignment based on the EM will generate better results for Ganon. We tested Ganon with the classification reassignment with the EM-algorithm on the 1M simulated reads generated by Mason. Ganon with reassignment achieved comparable F1 score as Centrifuger's (Figure S8 as shown below). Inspired by Ganon's improved performance, we also implemented a similar workflow to reassign a Centrifuger's multiple-classified read to its most abundant taxon. The abundance in this workflow is by directly counting the number of reads in a taxon or its subtree after initial classification. On this simulated data, Centrifuger with reassignment produced the best results, where it was both more sensitive and precise than Ganon with reassignment (Figure S8 as shown below). The implementation for Centrifuger with reassignment is at: https://github.com/mourisl/centrifuger_evaluations/tree/master/sim#centrifuger-reassign.

Figure S8. Sensitivity, precision and F1 score of Centrifuger and Ganon using LCA and reassignment based on taxa quantification at various taxonomy levels

However, we believe the reassignment based on the quantification results will introduce systematic biases to taxa with lower abundances. For example, suppose we have two species *A* and *B*, where *A*'s abundance is twice as much as *B*'s. Let's also suppose there are 100 reads, where 66 of them are from *A* and 34 are from *B*. The classifier finds that 2 reads are uniquely classified to *A*, 1 read is uniquely classified to *B*, and 97 reads are classified to both. Using EM algorithm, the abundance estimation will also find *A* is more abundant than *B*. As a result, the reassignment will push all the 97 reads to be uniquely assigned to *A*. The sensitivity at the species level before the reassignment is 3% with a precision at 100%. After reassignment, the sensitivity increases to 67% with the precision at 67%, which substantially improves the F1 score. But the downstream analysis based on the reassigned results will think there are 99 reads from *A* and 1 read from *B*, and this will create strong biases. Theoretically, because the reassignment favors more abundant taxa, the gain on sensitivity will be more than the loss on precision, yielding a better overall F1 score. We also created Ganon classification results of 100 reads mimicking the format from the option “---output-all”, where 2 reads were assigned to NZ_CP093546.1, 1 read were assigned to NZ_CP065615.1, and the remaining 97 reads were assigned to both. These two genomes were randomly selected, and served as name holders. We then ran “ganon reassign” and found that there were 99 reads uniquely assigned to NZ_CP093546.1 in the new “.one” file. The files used in this exploration of Ganon's reassignment results are at https://github.com/mourisl/centrifuger_evaluations/tree/master/other_scripts/ganon_reassign. These extreme examples demonstrate that though the reassignment based on taxonomic quantification can yield better classification results, this is at the expense of introducing systematic “underassignment” to the taxa with lower abundances. Therefore, we decide to use the LCA strategy, which reflects the classifier's accuracy on a single read and the results can be directly used in downstream analysis without creating strong biases.

In the revised manuscript, we add the experiments using reassignment strategy in the Methods section as following:

Ganon's default option for coalescing the taxonomy IDs of a multiple-classified read is to reassign the read to the taxonomy ID with the highest abundance inferred by the EM algorithm using the initial classifications. This approach substantially improves Ganon's classification sensitivity at lower taxonomy levels, leading to a higher F1 score. For example, in the Mason-generated simulated data, Ganon with reassignment's F1 score at the species level was 1.3% and 12.1% higher than Centrifuger's and Ganon using LCA's, respectively (Figure S8). We implemented a similar workflow to reassign the multiple-classified reads from Centrifuger. Specifically, we ran Centrifuger with “-k 5” so that the initial classification for a read could include up to five equally good classification results. We then calculated the abundance for each taxonomy ID in the taxonomy tree based on the number of reads classified to this taxonomy node and its subtree. For a multiple-classified read, we included its count starting from its LCA. Lastly, we reassigned the taxonomy ID with the highest abundance among the initial results to a read as the final classification result. When there were multiple highest-abundance taxonomy IDs for a read, we took their LCA as the final result. We observed this reassignment strategy without EM algorithm improved Centrifuger's classification results at lower taxonomy levels too, with the F1 score 2.7% higher than Ganon's reassignment results at the species level (Figure S8). However, the strategy of hard reassignment based on the taxonomic profiling result may result in systematic underestimations of taxa with lower abundances. Therefore, we continue to utilize the LCA strategy to process multiple-

Second round of review

Reviewer 1

The manuscript has improved quite a bit. As to the new results, specifically regarding the comparison with ganon (or ganon2) I have the following remarks which I would like see to be addressed:

1) ganon is configured by default to produce taxonomic profiling (prioritizes precision over sensitivity). Centifuger does not output tax. profiling but only tax. sequence classification (also called tax. binning), as mentioned in the discussion. The proper way to compare with ganon would be with using the "--binning" parameter, that optimizes for sequence classification. This is clear in ganon documentation and ganon2 pre-print (<https://pirovc.github.io/ganon/classification/>) but was not used.

2) ganon achieves best results with the EM-algorithm (which is the default) but was changed to LCA ("--multiple-matches lca"). Why?

3) About the speed comparison: Why not use mutithreading?

Reviewer 2

Authors have carefully revised the manuscript and answered all my comments.

Reviewer 3

Thank you Authors, I am glad to see that the manuscript's quality and the method description have improved following feedbacks and issues detected by reviewers.

Reviewer 4

I am very satisfied with the response of the authors and the substantial improvements made to the paper. The revisions have clearly addressed the initial concerns, enhancing the work's quality and impact significantly. I congratulate the authors for their diligent efforts and commendable work.

Reviewer 5

The revised manuscript addressed the majority of my comments. The only comment that remains is about a sizeable simulated test, which the Authors attempted to tackle with the concatenation of the existing tests. Could the Authors use the 50M read simulated datasets from Mende et al, Plos One, 2012? Could they report accuracy and runtime?

Dear Editor,

We thank the reviewers for their positive comments and valuable suggestions. We also appreciate the opportunity to submit the revision, and would like to highlight the following two major updates:

1. We reran Ganon for all the experiments on the short-read data sets with the option “-b” for taxonomic classification as suggested by Reviewer 1.
2. We compared the Ganon and Centrifuger using the strategy of reassigning the taxonomy IDs for multiple-classified reads based on taxonomic quantifications.

In the point-by-point response to the two reviewers, reviewer comments are in blue followed by our response in black. We hope that with all the reviewers’ questions addressed, our revised manuscript is now suitable for publication at Genome Biology.

Sincerely,
Li Song and Ben Langmead

--

Reviewer #1

The manuscript has improved quite a bit. As to the new results, specifically regarding the comparison with ganon (or ganon2) I have the following remarks which I would like see to be addressed:

1) ganon is configured by default to produce taxonomic profiling (prioritizes precision over sensitivity). Centrifuger does not output tax. profiling but only tax. sequence classification (also called tax. binning), as mentioned in the discussion. The proper way to compare with ganon would be with using the "--binning" parameter, that optimizes for sequence classification. This is clear in ganon documentation and ganon2 pre-print (<https://pirovc.github.io/ganon/classification/>) but was not used.

We appreciate the reviewer for pointing out the issue of running Ganon. Following the suggestions, we reran Ganon with the option “-b” (equivalent of --binning based on the documentation) and observed substantially better results from Ganon. Nevertheless, Centrifuger still performed significantly better at lower taxonomy levels, such as the species level and the genus level. The only case Ganon outperformed Centrifuger was the species-level classification accuracy in the experiment of removing true genomes from the database on the simulated data set. In this reply letter, we show the updated results for the evaluation on the simulated data set generated by Mason (Figure 3A as shown below), where Ganon is the second-best classifier and Centrifuger achieved 20% higher sensitivity at the species level. The other figures related to Ganon are also updated in the revised manuscript.

classified reads, where the results can be directly used for downstream analyses, including taxonomic profiling.

3) About the speed comparison: Why not use multithreading?

In our manuscript, we tested the classifiers with a single thread and multiple threads (8 threads). We compared the speed with a single thread on the 1M simulated reads generated by Mason. The multithreading experiment is on the concatenated CAMI data sets and the description in the revised results section is as following:

We concatenated the 10 samples into a large data set containing about 67 million read pairs to compare the speed of the classifiers running with eight threads. Kraken2 was the fastest method (finished in about 7min), followed by Centrifuge (42min), Centrifuger (1h35min), and Ganon (3h33min). With multithreading, Centrifuger was about 2.26 times slower than Centrifuge, reducing the threefold speed difference when running on a single thread. KMCP did not scale well and took more than 18 hours to finish.

--

Reviewer #5

The revised manuscript addressed the majority of my comments. The only comment that remains is about a sizeable simulated test, which the Authors attempted to tackle with the concatenation of the existing tests. Could the Authors use the 50M read simulated datasets from Mende et al, Plos One, 2012? Could they report accuracy and runtime?

We thank the reviewer for the suggestion of this simulated dataset. The manuscript “Assessment of Metagenomic Assembly Using Simulated Next Generation Sequencing Data” mentions the raw data is available at https://www.bork.embl.de/~mende/simulated_data/, but this website only stores some bacteria accession IDs as shown below. The “assemblies” folder is also empty. We explored the simulator, cMESSi, used in this study, but it has no documentation at <https://sourceforge.net/projects/cmessi/>, so reproducing the simulated data set will be beyond the scope of this project. We want to mention that each simulated data set from the CAMI2 study generated in 2019 contains a similar number of reads, so the average sensitivity and precision would correspond to the accuracy of a larger simulated data set.

Index of /~mende/simulated_data

Name	Last modified	Size	Description
 Parent Directory			-
 assemblies/	2011-12-06 12:52		-
 bacterial_data.txt	2011-05-24 17:59	37K	

Figure for Reviewer 5. The screenshot of https://www.bork.embl.de/~mende/simulated_data/